# Extreme event waves in marine ecosystems: an application to Mediterranean Sea surface chlorophyll

Valeria Di Biagio, Gianpiero Cossarini, Stefano Salon, Cosimo Solidoro

National Institute of Oceanography and Applied Geophysics - OGS

*Correspondence to*: Valeria Di Biagio (vdibiagio@inogs.it)

**Abstract.** We propose a new method to identify and characterise the occurrence of prolonged extreme events in marine ecosystems at the basin scale. There is growing interest in events that can affect ecosystem functions and services in a changing climate. Our method identifies extreme events as the peak occurrences over a predefined threshold (i.e., the 99th percentile) computed from a local time series and defines a series of extreme events that are connected over space and time as an extreme event wave (EEW). The main features of EEWs are then characterised by a set of novel indexes, which are referred to as initiation, extent, duration and strength. The indexes associated with the areas covered by each EEW were then statistically analysed to highlight the main features of the EEWs in the considered domain. We applied the method to a multidecadal series of winter-spring daily chlorophyll fields that was produced by a validated coupled hydrodynamic-biogeochemical model of the Mediterranean open sea ecosystem. This application allowed us to identify and characterise surface chlorophyll EEWs in the period from 1994-2012. Finally, a fuzzy classification of EEW indexes provided bio-regionalisation of the Mediterranean Sea based on the occurrence of chlorophyll EEWs with different regimes.

## 1 Introduction

Extreme events affecting the Earth system have been widely investigated in hydrology and atmospheric sciences (e.g., Delaunay, 1988; Katz, 1999; Luterbacher et al., 2004; Allan and Soden, 2008; Perkins and Alexander, 2013; Tramblay et al., 2014), also in connection with social sciences (Raymond et al., 2020). The study of extreme events in the ocean has mainly focused on sea levels (e.g., Zhang and Sheng, 2015), especially in relation to hydrology (e.g., Walsh et al., 2012), with recent works on extreme wave height (e.g., Hansom et al., 2014), current velocity (e.g., Green and Stigebrandt, 2003) and marine heat waves (e.g., Hobday et al., 2016, Galli et al., 2017). However, extreme events in marine biosphere properties (e.g., biogeochemical species concentrations) have received relatively little attention in recent years, despite the related heavy impacts on marine ecosystem functions and services (e.g., Zhang et al., 2010), with cascading effects at large scales on biogeochemical cycles (e.g., Doney, 2010).

Ocean warming, increase of atmospheric $CO_2$ and anthropogenic eutrophication are among the major stressors of marine ecosystems (Hoegh-Guldberg et al., 2018) and potential drivers of extreme events. Therefore, estimates of sea surface

temperature (SST), seawater pH, dissolved oxygen and the saturation state of $CaCO_3$ minerals are often used as probes to
monitor marine ecosystem health (e.g., Belkin, 2009; Andersson et al., 2011; Paulmier et al., 2011).
In particular, some studies on marine heat waves (Hobday et al., 2016), hypoxia events (e.g., Conley et al., 2009) and low
aragonite saturation states (Hauri et al., 2013) have identified extreme events at a given site by starting from the values of a
specific ecosystem variable that are above/below a certain threshold for a finite time duration. Although a common definition
of "extreme event" in this context is lacking, its main emerging features include (i) the intensity (i.e., the absolute difference
of the variable value with respect to the threshold), which is considered a large deviation from a reference ecological state; (ii)
the duration, which is considered a further stress factor on the ecosystem and is eventually combined with the intensity in an
overall "severity" index (as in Hauri et al., 2013); and (iii) the local characteristic, which is linked to the heterogeneity of the
ecosystem within the area of interest and/or due to sparse data sampling (e.g., from fixed stations). In fact, the spatial extension
of an extreme event is possibly evaluated in retrospect (e.g., Rabalais et al., 2002; Galli et al., 2017).
Our purpose is to design a general method that is able to not only capture the previously listed features of an "extreme event"
but also account for the persistence of the event within a certain impacted area and over a specific time duration (as Andreadis
et al., 2005 and Sheffield et al., 2009 proceeded in case of droughts) up to the basin scale.
The use of numerical models has been shown to be necessary to conduct such a study, which requires seamless and long
sampling times at high frequency at the basin scale. In fact, remote sensing observations are limited by cloud coverage and L4
data are based on filtered reconstructions (using climatology or the EOF method, as in Volpe et al., 2018), which can partly
mask the occurrence of extreme events. On the other hand, in situ measurements do not offer suitable spatial and temporal
sampling at the basin scale and can lack standardisation. In contrast, numerical models provide data with continuity at high
frequency in both time and space. Moreover, models also account for physical and biological processes that occur in marine
ecosystems in subsurface layers (e.g., vertical mixing, nutrient transport), allowing for a more complete reconstruction of the
dynamics of extreme events.
For our investigation, we used the dataset provided by the MITgcm-BFM hydrodynamic-biogeochemical model of the
Mediterranean Sea ecosystem at 1/12° horizontal resolution (Di Biagio et al., 2019).
In particular, we applied the proposed method to the surface chlorophyll concentration, an essential ocean variable (EOV, e.g.,
Muller-Karger et al. 2018) that is representative of the marine ecosystem state and evolution. This application allowed us to
identify the extreme events of surface chlorophyll, which can correspond to both phytoplankton blooms (Desmit et al., 2018)
and positive anomalies with values too low to be actually considered "blooms" (e.g., in the Levantine Sea). In fact, due to the
general oligotrophy of the basin, marked increases in phytoplankton chlorophyll are strictly considered "blooms" in only some
regions (north-western Mediterranean Sea, Alboran Sea, Catalan-northern Balearic area, isolated coastal areas near some river
mouths, see Siokou-Frangou et al., 2010). Our method is instead formulated to identify extreme values of chlorophyll as peaks
over a threshold defined in the time series of all basin points, i.e., from local and statistical perspectives. We focused our
investigation on the open sea domain, thus avoiding the areas that are directly affected by bottom and riverine dynamics (e.g.,
Oubelkheir et al., 2014).

The article is structured as follows. The Material and methods section (Sect. 2) is divided into two parts, presenting the method to identify the extreme events and the model-derived dataset that was used (Sect. 2.1 and Sect. 2.2, respectively). Section 2.1 consists of three parts: the identification of local extreme events and extreme event waves (EEWs, Sect. 2.1.1), characterisation (Sect. 2.1.2) and classification (Sect. 2.1.3). The results are presented in Sect. 3 and Appendix A (examples of identification of extreme events of surface chlorophyll). In particular, Sect. 3.1 shows a chlorophyll EEW, identified and characterised following the proposed method, with further analyses in terms of its internal physical and biogeochemical dynamics given in the Appendix B. Section 3.2 presents the classification of all the modelled chlorophyll EEWs in the Mediterranean Sea from 1994-2012, including a sensitivity test for the local thresholds. Section 4 includes a discussion about the proposed method and the main results, and it is followed by the conclusions in Sect. 5.

## 2 Material and methods

### 2.1 The method for the spatio-temporal investigation of extreme events

The method illustrated here allows us to identify, characterise and classify the "extreme event wave" in marine ecosystems, accounting for their time duration, intensity and spatial extension, and with reference to any ecosystem variable $C(x,y,z,t)$. Hereafter, we refer to a daily sampling of the variable time series and a two-dimensional variable $C(x,y,t)$ for simplicity. However, the method can be easily extended to any regular time discretization and to the 3D spatial case with few modifications in the definition of the indexes, as discussed in Sect. 4.

### 2.1.1 Identification

We define "extreme events" as the occurrences of values $C(x,y,t)$ that are higher than a reference percentile threshold (e.g., Asch et al., 2019), computed over the whole time series of the variable. In particular, we search for the peaks over threshold (POTs), with the threshold referring to the 99th percentile of the time series. Extreme events are thus represented by the highest values (i.e., top 1%) of the variable distribution observed in the $(x,y)$ point. These events are "rare" events and are selected from the total records independently from their distribution over the years.

Then, we identify a "wave" of extreme events, or "extreme event wave" (EEW), as a set of extreme events that are connected in space and time. Thus, an EEW tracks anomalous events that are not merely local but that co-occur in more than one grid point and possibly are transported in space and evolve over time.

Operationally, the spatial and temporal occurrence of all the POTs can be mapped on a binary 3D matrix, representing the (2D map x day) flags of the extreme events, equal to 1 for the $(x,y,t)$ points of POT occurrence and 0 for the points without POT occurrence. EEWs are then defined as sets of POT occurrences that are "the closest neighbours" in space and in time.

The EEW definition is thus a filter for the spatio-temporal dataset: it allows us to identify single events that affect a portion of the domain for a certain time period, in which all the involved points display extreme values of the selected variable. In the

EEW, the spatial contiguity of the points with variable values above their own threshold at the same time is a further request
(e.g., Andreadis et al., 2005), which adds to the temporal contiguity typical of the definition of local extreme events (e.g.,
Hobday et al., 2016).

### 2.1.2 Characterisation

We introduce the characterisation of EEWs based on two kinds of metrics: spatio-temporal indexes (sketched in Fig. 1) and
strength indexes (sketched in Fig. 2).
The spatio-temporal indexes, which are used to localise and describe an EEW in space and time (green shape in Fig. 1), are
the following:
● the *initiation*, as the first day when at least one POT belonging to the EEW occurs;
● the *duration T* (yellow arrow in Fig. 1), as the time interval in which there are POTs included in the EEW. This metric
is labelled by the maximum temporal difference between two POTs of the EEW, in *day* units;
● the *area A* (grey area in Fig. 1), as the union of all the surface grid cells housing the POTs included in the EEW. This
metric is labelled by the sum of these cell areas, measured in $km^2$;
● the *width W*, as the measure of the spatio-temporal region occupied by the EEW. This metric is computed as the sum,
over the grid points covered by $A$, of the spatio-temporal regions identified by the grid point area as the base and the
total time interval of POTs of the EEW referred to that grid point as the height. This metric is measured in units of
$km^2 \times day$;
● the *uniformity U*, as the ratio between the width $W$ and the spatio-temporal region defined by the prism with $A$ as the
base and $T$ as the height:
$$U = \frac{W}{AT} \qquad\qquad (2.1)$$
This metric represents the percentage of the prism that is occupied by the EEW and quantifies how persistent (on
average) the EEW is in the single grid point belonging to $A$.
We excluded both EEWs with duration $T<2$ days (e.g., Asch et al., 2019) to neglect possible transient spikes and EEWs with
area $A < 4\Delta x \times 4\Delta y$ (with $\Delta x$ and $\Delta y$ grid spacing in the zonal and meridional directions, respectively) since the estimated
factor between the effective resolution and grid spacing of the numerical models is 4 or more (Grasso, 2000).
The strength indexes of an EEW can be defined starting from some quantities that are introduced locally (sketched in the top
right box of Fig. 1). That is, considering the time series at each grid point *(x,y)*, the *j−th* POT included in the EEW is
characterised by the value $C_j(x,y)$ of the ecosystem variable and by the intensity $I_j(x,y)$ above the threshold $p99(x,y)$ computed
on the time series:
$$I_j(x, y) = C_j(x, y) - p99(x, y). \qquad\qquad (2.2)$$
Given the set $J_{(x,y)}$ of all the occurrence indexes *j* of the POTs that refer to the specific grid point *(x,y)* and are included in the
EEW, we can define the following strength indexes:
● the *severity S*, as the total "mass" of the variable characterising the EEW, which is computed as the sum over the grid
points covered by *A* of the local sum $M(x,y)$ of the "masses" supplied by the POTs included in the EEW:
$$S = \sum_{(x,y)\in A} \quad M(x,y) = \sum_{(x,y)\in A} \quad \sum_{j\in J_{(x,y)}} \quad C_j(x,y) \tag{2.3}$$
The severity is represented in a simplified way in Fig. 2a.
● the *excess E*, as the total intensity above the "threshold" (i.e., the locus of points of the local thresholds, *P99*),
associated with the EEW. Its formulation is analogous to eq. (2.3) but referred to $I_j(x,y)$ rather than $C_j(x,y)$:
$$E = \sum_{(x,y)\in A} \quad I(x,y) = \sum_{(x,y)\in A} \quad \sum_{j\in J_{(x,y)}} \quad I_j(x,y) \tag{2.4}$$
Additionally, the excess is represented in a simplified way in Fig. 2a.
Depending on the ecosystem variable under investigation, eqs. (2.3) and (2.4) may require multiplication by the cell
volume or by the cell area in the inner summation to provide a consistent unit of measurement (e.g., if $C_j(x,y)$ is a
concentration, it should be multiplied by the cell volume $V(x,y)$ to obtain an actual mass).
● the *mean severity* $< S >$, as the ratio between the severity and the width *W* of the EEW:
$$\langle S \rangle = \frac{S}{W} \tag{2.5}$$
● the *anomaly,* as the ratio between the excess and the severity:
$$AN = \frac{E}{S} \tag{2.6}$$
This index represents the percentage of the excess in the severity of the EEW. The index, which is adimensional, is
sketched in Fig. 2b for two different EEWs, which have the same severity but different excess and, thus, anomaly.
Since the locus of points of the thresholds of the second EEW is lower, this EEW has a higher anomaly value than
the first one and has a larger impact on the ecosystem.
**2.1.3 Classification**
The EEWs introduced in our formulation are identified starting from the local 99th percentile thresholds of the variable time
series. The concept of "extreme" adopted in this work is related to the local characteristics of the marine ecosystem, which can
be largely heterogeneous across the domain. Here, we propose a classification of EEWs suitable to highlight the main features
of EEWs in the considered spatial domain.
For each index defined in Sect. 2.1.2, the values obtained for each EEW can be associated with all the points belonging to the
*A* areas (Fig. 1), and then, a mean map over of the considered spatial domain can be obtained by averaging all the values of
the related index point by point. Finally, fuzzy clustering analysis (Bezdek et al., 1984) can be conducted on the mean maps
of all the indexes. In this way, different bio-regions of EEWs can be identified, depending on the relative weight of the indexes
under consideration, and specific regimes of EEW can therefore be highlighted.

**2.2 Data: Mediterranean Sea surface chlorophyll by MITgcm-BFM**

We used the results of the 1994-2012 hindcast simulation discussed in Di Biagio et al. (2019) and produced by the MIT general circulation model (MITgcm, Marshall et al., 1997), coupled with the biogeochemical flux model (BFM, Vichi et al., 2015) following the online scheme described in Cossarini et al. (2017). The configuration in use has a horizontal resolution of 1/12°, with 75 unevenly spaced vertical levels. The atmospheric fields used to force the simulation come from a 12 km horizontal resolution regional downscaling of ERA-Interim reanalysis (Llasses et al., 2016; Reale et al., 2017) and drive the simulation every 3 hours.

In particular, we used the daily chlorophyll concentration computed at the surface (i.e., averaged on the first 10 $m$), restricting our investigation to the January-May period. We considered only grid points with depths greater than 200 $m$, which are identified as the "open sea". With this spatial constraint, we neglected both the coastal points, which are directly affected by river nutrient discharge, and the points where interactions with the sea bottom occur within the euphotic layer, since modelled variables in these regions are possibly affected by high uncertainties (Di Biagio et al., 2019).

The chosen MITgcm-BFM simulation has the characteristics required by the extreme events analysis: it is seamless, it provides high frequency patterns (as shown by wavelet analysis in Di Biagio et al., 2019), it lacks spurious leaps in the ecosystem state (that might occur in case of filter data assimilation process, Teruzzi et al., 2014), and it reproduces the heterogeneity of the marine ecosystem across the basin and its main biogeochemical properties.

**3 Results**

**3.1 Chlorophyll EEWs: identification and characterisation**

We applied the method illustrated in Sect. 2 to the surface chlorophyll concentration, so that $C(x,y,t) \equiv chl(x,y,t)$, measured in $mg\ m^{-3}$, with daily samplings provided by the simulated Mediterranean Sea biogeochemistry in the 1994-2012 period (Sect. 2.2). The local "extreme events" for this application thus correspond to the top 1% values of the surface chlorophyll distribution in each grid point (Sect. 2.1.1). Examples of extreme events of surface chlorophyll in Mediterranean Sea are reported for selected sites (Fig. 3) in Appendix A, showing that the different sites have different p99 and that not all annual maxima correspond to the definition of local "extreme events".

From the ecosystem point of view, the most suitable indexes to describe the phenomenology of the surface chlorophyll EEWs (i.e., continuous and prolonged "waves" of extreme events) are the mean severity $<S>$, the anomaly $AN$, the duration $T$ and the uniformity $U$ (Sect. 2.1.2). In fact, when considering the surface chlorophyll as a proxy for surface phytoplankton biomass (e.g., Boyce et al., 2010), the mean severity provides the mean amount of biomass supplied by the EEW to the surface layer in 1 day over a unit area of 1 $km^2$ and is expressed in $kg\ km^{-2}\ day^{-1}$. The anomaly index instead represents the anomalously high amount of chlorophyll with respect to the history of the local ecosystem. The duration $T$ measures the overall ongoing impacts on the marine ecosystem, which are considered in this study as responses of ecosystem processes and status to an

excess of phytoplankton biomass. On the other hand, the uniformity index quantifies the local persistence of the chlorophyll EEW on the points included in area $A$. In fact, with constant values of $A$ and $T$, an EEW with higher $U$ will affect the single unit of $A$ for longer times, with higher potential ecological consequences on the ecosystem unit.

Considering the temporal extension of the simulation (approximately equal to 7000 days), the computation of the 99th percentile over the time series gives a value equal to 70 for the number of POTs in each grid point. The mapping of the 99th percentile threshold values computed at each grid point throughout the basin (Fig. 3) indicates that grid points that are spatially close exhibit small differences in their threshold values and that different patterns are recognisable in the basin. Hereafter, we use the abbreviations indicated in Fig. 3 to refer to different Mediterranean Sea regions.

The total number of surface chlorophyll EEWs, which were identified by applying the definition (Sect. 2.1.1) and the further requests on the $A$ and $T$ indexes (Sect 2.1.2) in the investigated period, was 947. We show in Fig. 4 an example of a detected EEW, which is represented in the spatio-temporal domain and compared with remote sensing data (http://marine.copernicus.eu/, product OCEANCOLOUR_MED_CHL_L3_REP_OBSERVATIONS_009_073, Volpe et al., 2019). The values of the spatio-temporal and strength indexes computed for this EEW are summarised in Tab. 1. The EEW occurred in the Gulf of Lion (region NWM) during early spring (15th-31st March 2005). Both the model and satellite data show patterns of high values of chlorophyll in the period of EEW occurrence (second and first columns of Fig. 4, respectively). Strong increases in chlorophyll in the Gulf of Lion and the Ligurian Sea are recognisable on the satellite maps starting from 20th March, after a period of very low chlorophyll concentration (even lower than 0.05 $mg\ m^{-3}$, not shown). Although the model uses a spatial resolution (approximately equal to 7 $km$) that is coarser than the satellite resolution (1 $km$), it is able to capture a surface signature typical of deep convection dynamics (second column, compared with the first column, on 20th March). However, the comparison between the model and satellite data points out that the impact of cloud coverage on remote sensing measurements is a limiting factor for the reconstruction of the spatio-temporal dynamics of chlorophyll extreme events. The comparison of the modelled chlorophyll maps (second column) with the patterns of the daily area $A$ of the EEW (third column) shows that the EEW patch actually includes points with noticeably high chlorophyll values in the region. Nevertheless, $A$ also contains points with chlorophyll values that are low on the same absolute scale, yet higher than the local 99th percentile thresholds (as ensured by our procedure). Moreover, the EEW patches appear to be advected by the current velocity field (third column) and to follow both convection weakening (see plots in consecutive panels) and the patches of high nutrient concentrations in the previous days (by comparison with the right panel referred to the day before). The high values of chlorophyll recorded in the Ligurian Sea on 20th March are associated with a separate EEW (not shown).

From Tab. 1, we quantify a mean severity equal to 1.389 $kg\ km^{-2}\ day^{-1}$ and an anomaly index equal to 0.205% for this EEW. In fact, this EEW was the most severe and the sixth most anomalous in all of the EEWs identified in the Mediterranean domain, as reproduced by our simulation. This result indicates that the large amount of chlorophyll supplied by this EEW was also considerably high throughout the history of the impacted local ecosystem. Moreover, even if the overall duration of the EEW was 17 days, each unit area was actually affected for only approximately 3 days ($U\ T \approx 3\ days$). This result means that the EEW spread out in space and time with an articulated shape, as shown in Fig. 4.

In the Appendix B, further analysis conducted on three points that are located internally, externally and on the border of the
EEW area showed that the EEW identification actually takes into account all and only the relevant information associated with
it; thus, that the proposed method acts like a filter to properly circumscribe the extreme events in space and time. In fact, this
specific EEW captured the dynamics of the exceptionally intense bloom observed in the NWM in 2005 (Estrada et al., 2014;
Mayot et al., 2016), which was triggered by very strong vertical mixing (and deep convection, in the internal point), followed
by the restoration of stratification.

**3.2 Classification in the Mediterranean Sea**

This section shows the results of the basin-scale classification of all the surface chlorophyll EEWs identified from 1994-2012
(Sects. 3.2.1 and 3.2.2) by means of the spatial spreading of the values of the indexes on the areas covered by the EEWs and
the subsequent clusterisation of the mean maps of the indexes (as explained in Sect. 2.1.3). This section also displays the results
of a sensitivity test of the EEW indexes, which were averaged over the outcome clusters, to different thresholds computed on
the local time series (Sect. 3.2.3).

**3.2.1 Mean maps of the indexes**

Figure 5 displays the total number of EEWs that occurred in each point of the Mediterranean domain (Fig. 5a) and the mean
values of the EEW indexes, which were computed as the mean of the indexes of all the EEWs that involved that point (Figs.
5b-f).
Since some Mediterranean areas show more than one EEW per year (as can be inferred from Fig. 5a), the initiation time in
each grid point and year was associated with the most severe EEW of that year. We found that the most severe chlorophyll
EEWs occurred mainly in the winter months in the central and southern open sea parts of the basin and later in the early spring
period in NWM, central ALB, northern ION, ADS, AEG and the Rhodes Gyre (Fig. 5b). The duration of the chlorophyll EEWs
reached 90 days in the southern part of the eastern basin, whereas it decreased to 30 days in NWM and the Rhodes Gyre and
to approximately 15 days in ALB and ADS (Fig. 5c). Long duration was typically associated with low uniformity (e.g.,
southern ION and LEV areas), while EEWs with high uniformity were found in ADS and ALB (Fig. 5d). The western
Mediterranean displayed the EEWs with the highest mean severity, which were associated with the highest produced biomass
in ALB and NWM (Fig. 5e). Nevertheless, the regions with the highest values of anomaly occurred in the eastern ADS and
the northern Ionian Sea (Fig. 5f), despite their values of severity being approximately 50% lower than the values displayed in
ALB and NWM.

**3.2.2 Clusterisation of the indexes**

Figure 6 displays the clusterisation provided by fuzzy k-means analysis (Bezdek et al., 1984) conducted on the maps of the
main indexes (i.e., duration, mean severity, uniformity and anomaly, Fig. 5) by adopting a fuzziness parameter equal to 2.
Seven Mediterranean Sea regions with similar chlorophyll EEWs phenomenology were identified by the maximum

membership values and are indicated by different colours in Fig. 6. To evaluate the robustness of the clusterisation, we also computed the "confusion index", i.e., one minus the difference between the dominant and subdominant memberships for each point (Burrough et al., 1997). We obtained a value less than 0.7 (i.e., limit for "high confusion" condition) for the largest part of the domain. Values higher than 0.7 are shown in only patchy and limited areas (e.g., part of the southern Adriatic Sea, Fig. 6). Moreover, we computed the mean and standard deviation of the indexes within the seven clusters to quantify the mean impact of the EEWs on the related ecosystem (Tab. 2). Finally, we estimated the trends of duration, mean severity, uniformity and anomaly in the simulated period (1994-2012), applying the Theil-Sen method (Theil, 1950 and Sen, 1968) to the annual means of the indexes computed on the points included within the clusters. The red (blue) colour in Tab. 2 indicates an annual increase (decrease) higher than 1%.

In Fig. 6, cluster #7, which covers NWM and eastern ALB, displays EEWs with durations of 29 days, with the highest values of mean severity (approximately equal to 1 $kg\ km^{-2}day^{-1}$), along with high anomaly and intermediate uniformity with respect to the other clusters (Tab. 2). The EEWs with the highest uniformity ($U \simeq 0.28$) and the shortest duration ($T=26\ days$) were identified in the areas of cluster #6, i.e., ADS, ALB, the coastal areas of the southwest Mediterranean and some spotted areas in the north Ionian Sea, TYR, Sicily Strait, the Rhodes Gyre and AEG. These areas display intermediate severity and low anomaly. Relatively high uniformity ($U \simeq 0.22$) also characterises cluster #5, in northern and eastern LEV, AEG and the southern coastal areas of ION, with intermediate duration and low severity and anomaly values. Cluster #4 represents the EEWs with the longest duration ($T=63\ days$) and lowest uniformity ($U \simeq 0.14$), with low severity but relatively high anomaly. Cluster #1, corresponding to most of the North Ionian Sea, eastern ADS and spotted areas in the western central Mediterranean, displays the EEWs with the highest anomaly (i.e., $AN \simeq 0.135$), along with intermediate values of the other indexes. Clusters #3 and #2 display very similar intermediate values of uniformity and anomaly, but the EEWs in cluster #2 exhibit lower severity and longer duration than cluster #3.

Throughout the simulated period, the western Mediterranean (except ALB), which was identified by clusters #3 and #7, did not display any significant trend (Tab. 2). On the other hand, the duration of EEWs in the eastern sub-basin and in ALB increased, whereas the uniformity of EEWs in ION and south-eastern LEV (i.e., clusters #1, #2 and #4) decreased. A significant increase in the anomaly was recorded in the eastern basin, except for ADS and spotted areas in AEG and the Rhodes Gyre (i.e., clusters #1, #2, #4, #5).

### 3.2.3 Sensitivity to the threshold

We conducted a sensitivity test of the method to two different thresholds computed over the time series in each grid point. We repeated the steps of the method (Sects. 2.1.1-2.1.3) up to obtaining the mean maps of the indexes on the Mediterranean domain for the 98th and 99.5th percentile thresholds. Then, we spatially averaged the values of the indexes within the seven clusters of Fig. 6 and finally we computed the total means by averaging the means of the seven clusters. Figure 7 shows the results compared with the 99th percentile (i.e., p99) reference threshold.

The duration and anomaly indexes show decreasing values for increasing thresholds. In contrast, the mean severity and uniformity display increasing values. The relative cluster ranks are generally preserved. Moreover, clusters #4, #6, #7 and #1 maintain the highest values of duration, uniformity, mean severity and anomaly, respectively, for all the selected thresholds, except in the case of the overall mean anomaly for the 98th percentile, in which anomaly values of cluster #7 overcome those of cluster #1 (Fig. 7).

**4 Discussion**

In this work, we propose a new method to identify and characterise extreme event waves in marine ecosystems. The method is then exemplified by a first application to surface chlorophyll in Mediterranean open sea areas, with specific reference to the winter-spring period.

One of the key points of this method is the definition of an "extreme event". In fact, the spatial extension of extreme events is scarcely addressed in the literature, although it can be an important ecosystem indicator, e.g., to predict a possible recovery of an ecosystem (O'Neill, 1998; Thrush et al., 2005), and it is sometimes estimated a posteriori (Rabalais et al., 2002). In contrast, the spatial contiguity of local extreme events has been evaluated here in addition to the temporal contiguity, following Andreadis et al. (2005). In this way, the definition of the extreme event from a time series in a grid point was extended to define the EEW, which covers an extended area for a certain time duration. Consequently, the metrics necessary to characterise and classify biogeochemical extreme events that can be introduced for a time series at specific sites (e.g., in Hauri et al., 2013; Hobday et al., 2016; Asch et al., 2019; Salgado-Hernanz et al., 2019) have been further developed to describe the shape and strength of EEWs and provide meaningful insights into related biogeochemical phenomenology.

In our specific application, it is noteworthy to specify that the top 1% values of surface chlorophyll (i.e., extreme events, as defined in Sect. 2.1.1) do not necessarily correspond to "blooms" (Siokou-Frangou, 2010) since extreme events are identified in all points of the domain, including oligotrophic areas. Moreover, the top 1% values of chlorophyll are not necessarily distributed in a regular way over the years due to the inter-annual variability of the chlorophyll time series (Figs. A.1-A.3). Our method is able to characterise the intensity and regularity of extreme events in retrospect by means of the mean severity and anomaly indexes computed on the EEWs.

In particular, the mean severity index associated with a chlorophyll EEW can be interpreted as the mean amount of biomass supplied daily to the sea surface over a unit area and could be used as an indicator of eutrophication (Gohin et al., 2008; Ferreira et al., 2011) and food availability for secondary production (Calbet and Agustí, 1999; Ware and Thomson, 2005). The map of the mean severity index obtained for the 1994-2012 period (Fig. 5e) revealed the heterogeneity of the chlorophyll EEWs in the Mediterranean Sea and is in good agreement with the spatial patterns of the chlorophyll amplitude index shown in Salgado-Hernanz et al. (2019), with the highest values recorded in ALB and NWM.

The anomaly feature, which corresponds to the case when the supplied biomass is much higher than usual for a certain area, can instead be ascribed to the inter-annual variability of the extreme events of surface chlorophyll (as in Mayot et al., 2016).

As an example, the reconstruction of the most severe and highly anomalous EEW that occurred in NWM in 2005 showed that the main variables exhibited significant deviations from the climatological values (Figs. B.2-B.3). Nevertheless, high anomaly values do not necessarily correspond to high values of mean severity. In fact, the highest anomaly values are in the northern ION and eastern ADS, which display relatively low values of mean severity (see Fig. 5f, compared with Fig. 5e). The anomaly highlights the episodic occurrence of the chlorophyll EEWs in some areas, such as the northern ION, where the surface chlorophyll values exceed the local p99 thresholds, approximately equal to 0.6 $mg\ m^{-3}$ (Fig. 3), in only some years (e.g., 1999, 2002, 2010), reaching values also higher than twice the p99 thresholds (as shown in the selected ION site time series in Fig. A.2).

On the other hand, the uniformity feature, i.e., the persistence of a chlorophyll EEW in a certain area, can be linked to specific spatial constraints that circumscribe the EEW. In particular, the circulation structure can play an important role in providing the high values of uniformity in Fig. 5d. In fact, permanent cyclonic gyres in ADS (which also impose a topological constraint) and northern LEV (i.e., the Rhodes Gyre; Pinardi et al., 2015) potentially support a major vertical transport of nutrients and, consequently, increased biomass values (Siokou-Frangou et al., 2010). Moreover, regular upwelling near the southern coast of Sicily can explain the high uniformity values in the Sicily Strait (e.g., Patti et al., 2010). Finally, other spotted areas with high uniformity in the ALB, SWW and TYR areas are characterised by semi-permanent mesoscale structures that are associated with the inflow of Atlantic water (Navarro et al., 2011), eddies originating from the Algerian Current (Morán et al., 2001) and dynamics of the northern TYR gyre (Artale et al., 1994; Marullo et al., 1994; Marchese et al., 2014), respectively.

The fuzzy k-means analysis in this study used mean severity, anomaly, duration and uniformity to classify the EEWs in the Mediterranean Sea. The initiation index was excluded from the computation since there are areas of the basin showing more than one EEW per year per grid point (Fig. 5a). However, as a general characterisation of EEW occurrence, the initiation index showed a south-north gradient from winter to early spring for the most severe EEWs in the open sea areas of the Mediterranean (Fig. 5b), which is in agreement with the phenology of surface chlorophyll in the Mediterranean Sea reported by D'Ortenzio and Ribera d'Alcalà (2009).

We obtained robust clusterisation, with only some areas of the domain showing a high confusion index (Fig. 6). This subdivision of the Mediterranean Sea displays several similarities to previous Mediterranean bio-regionalisations (D'Ortenzio and Ribera D'Alcalà, 2009; Lazzari et al., 2012; Ayata et al., 2018; Salon et al., 2019), indicating that the four indexes are meaningful in characterising the heterogeneity of the basin.

In particular, cluster #7, corresponding roughly to the north-western area (as the "Bloom" region in D'Ortenzio and Ribera d'Alcalà, 2009; NWM in Lazzari et al., 2012), has been associated with the highest mean severity (Tab. 2). A decreasing gradient of the mean severity is observed toward the eastern Mediterranean areas (clusters #3 and #6 showed higher values of mean severity than clusters #1, #2, #4 and #5), which is in agreement with the west-to-east oligotrophication gradient (e.g., D'Ortenzio and Ribera d'Alcalà, 2009; Colella et al., 2016) and the gradient of surface chlorophyll maxima (i.e., map of amplitude index by Salgado-Hernanz et al., 2019). Moreover, this cluster is also characterised by a very high anomaly content, highlighting that EEWs occurring in cluster #7 can occasionally supply a very substantial amount of chlorophyll, as in the case

of the already mentioned EEW that occurred in 2005, which developed after a deep convection event (see Appendix B). This interpretation is in agreement with that referring to the "High Bloom" regime, which takes the place of the "Bloom" regime in some years in the NWM area (Mayot et al., 2016).

Cluster #1 identified the regime with the highest anomaly (i.e., high inter-annual variability of the EEWs) and a decoupling between mean severity and anomaly. This regime in the Mediterranean Sea is found in the northern ION and eastern ADS (Fig. 6).

Both clusters #5 and #6 are associated with high uniformity and low anomaly, i.e., EEWs that are well localised in space and that regularly occur over the years. Nevertheless, cluster #6 is characterised by EEWs that have higher biomass content (i.e., more severe) and that expire more quickly (i.e., shorter) than those in cluster #5. In this way, ALB, coastal SWW, ADS and the central part of the Rhodes Gyre (i.e., cluster #6, Fig. 6) are differentiated by the south-western ION, AEG, and outer part of the Rhodes Gyre (i.e., cluster #5).

Cluster #4 displays the longest EEWs with the lowest uniformity, which are rare (not shown) and have relatively high anomaly and low severity. This typology of EEWs identifies a regime of spatially diffuse extreme events of surface chlorophyll whose values do not markedly differ from the chlorophyll means in the concerned area. This cluster covers a large part of the south-eastern Mediterranean Sea (Fig. 6) and is crossed by the Atlantic-Ionian Stream and the Cretan Passage Southern Current (Pinardi et al., 2015). We ascribe the very low uniformity and the high overall duration of these EEWs to the transport and spreading of chlorophyll along the meanders of these currents (not shown).

Finally, clusters #2 and #3 display in-between conditions with respect to others, since the values of all the indexes are intermediate, except the mean severity, which is very low in cluster #2 and relatively high in cluster #3. In the Mediterranean basin, this result corresponds to the decreasing gradient in the severity between the central part of the western part (i.e., cluster #3) and of the eastern (cluster #2) Mediterranean Sea (Fig. 6).

The obtained clusters (Fig. 6) were also used to evaluate the long-term evolution of ecosystem phenomenology. Our results did not show any increase in the intensity (i.e., in the severity index) of surface chlorophyll EEWs in any clusters over the period from 1994-2012 (Tab. 2). This result is in agreement with the estimations of trends in the amplitude index by Salgado-Hernanz et al. (2019), except in NWM. Moreover, no significant trend (defined here as an annual variation in an index that was higher than 1%) was estimated for any of the four indexes in the central and north-western sub-basin. In contrast, the eastern Mediterranean and ALB showed trends in the duration, uniformity and anomaly of chlorophyll EEWs. In particular, positive trends of duration found in areas with very uniform EEWs suggest a persistence of extreme events of chlorophyll that has prolonged over time. On the other hand, positive trends of duration and anomaly, along with low values of uniformity (with also negative trends), denote an increase in EEWs with articulated shapes in the areas with low productivity. The trends recognised in the eastern Mediterranean Sea suggest a possible increased tendency of this sub-basin to changes in the identified regimes, despite the productivity being lower than that in the western Mediterranean. This result is one of the features that could emerge only because we accounted for the local thresholds in the identification of the EEWs (Sect. 2.1.1).

The choice of the local percentile threshold (Sect. 2.1.1) is a critical parameter of our method. In our case, this threshold was computed as the 99th percentile on the surface chlorophyll time series in each grid point. A priori, the choice of a higher (lower) threshold corresponds to a definition of an "extreme" value that is narrower (broader) than the reference value. As shown in Fig. 7, the choice of higher thresholds increases the mean severity index, since it is computed on local values that are higher. In contrast, both the anomaly and the duration indexes decrease at increased thresholds because of the occurrence of local POTs over a smaller number of days (i.e., shorter duration) and the decreased detectability of the inter-annual variability (i.e., lower anomaly). The increase in the uniformity index is due to the promotion of grid points with more similar values of high local thresholds (i.e., closer in space, see Fig. 3). However, uniformity shows lower sensitivity to the threshold than the other indexes because of the occurrence of POTs in a few grid points with thin spatial connectivities extending over great distances (as discussed in Sheffield et al., 2009). In this case, a further sensitivity analysis over the area covered by the EEWs could be envisaged to identify a minimum area threshold (stricter than the $4\Delta x \times 4\Delta y$ constraint introduced in Sect. 2.1.2) to better characterise the uniformity.

Overall, Fig. 7 shows that the identification of the clusters with the highest index values was generally maintained in the case of both higher and lower thresholds, confirming that the main regimes of chlorophyll EEW were identified in a robust way. Since different variables of interest could highlight a different sensitivity of the indexes, we believe that conducting analyses with different local thresholds could help to identify the specificities of the phenomenology underlying the extreme events.

In fact, we have applied this method to surface chlorophyll, as one of the most representative and investigated variables of the marine ecosystem, which potentially influences ecosystem function (e.g., food web and carbon fluxes). However, our method can be applied to any ecosystem variable, including other phytoplankton variables (e.g., HAB-like phytoplankton groups, Vila and Masó, 2005), temperature, oxygen and fluxes (e.g., carbon fluxes at the ocean-atmosphere interface, von Schuckmann et al., 2018). The $C(x,y,t)$ variable can be defined at the surface, the sea bottom (e.g., oxygen minimum or oxygen deficiency, OSPAR, 2013; Ciavatta et al., 2016), and specific surfaces in the ocean interior (e.g., deep chlorophyll maximum, Lavigne et al., 2015; Salon et al., 2019), or it can be vertically integrated (e.g., integrated chlorophyll, which accounts for subsurface growth of phytoplankton). In some cases, the selected variable may require multiplication by the cell volume (e.g., if the variable is a concentration) or by the cell area (e.g., for surface fluxes and vertically integrated variables) in eq. (2.3) and eq. (2.4) to provide a consistent and meaningful definition of the severity and the excess indexes, respectively. Moreover, the formulation illustrated in Sect. 2.1 could be extended to the full 4D case (i.e., to variables $C(x,y,z,t)$) by adding the vertical dimension to the definition of a local extreme event (i.e., the POTs could be defined in each point in a 3D space) and an EEW (i.e., as 3D spatial volume connected in time). The spatio-temporal indexes would then refer to the spatial volume instead of area $A$ and to the 4D width and prism associated with the definition of uniformity. The 4D formulation could be applied to investigate, for instance, marine hypoxia by identifying volumes in time with extremely low values of oxygen. In this case, a proper threshold for local extreme events might also be defined by a constant value in space in connection with the impacts on benthic fauna and fish species, which have physiological limits (e.g., Rabalais et al., 2002, Vaquer-Sunyer and Duarte, 2008). Strength indexes would be modified and defined as vertical profiles, instead of scalar metrics, to show the intensity and depth

of the bottom ecosystem stress. Therefore, the novelty of our method (i.e., the temporal and spatial connection of extreme
events) would allow us to compute the extension of the (connected) spatial 3D volumes under hypoxic conditions to estimate
the probability of fish survival, which can be enhanced by swimming (avoidance) behaviour (Rose et al., 2017).
Finally, our method of EEW identification, characterisation and classification can also be applied to extreme events that are
defined starting from seasonally varying thresholds (e.g., in Hobday et al. 2016). In this case, "extreme events" would
correspond to the highest anomalies recorded with respect to the climatological seasonal cycle of the variable and generally
not to the highest values of the variable recorded throughout the time series (as in our case of temporally fixed threshold). Such
an application would allow us to investigate different kinds of scientific questions, such as chlorophyll anomalies in summer.
**5 Conclusions**
The present study provides a methodology to describe statistically extreme events in a marine basin-scale ecosystem and is
supported by an ecological interpretation.
A key point of the method is the request of contiguity in both time and space of the peaks over the local threshold of the
ecosystem variable. This constraint allowed us to define individual events as extreme event waves (EEWs) occurring in
localised spatio-temporal regions. In particular, we accounted for the contiguity of the local extreme events, which is an aspect
that has been rarely considered in the literature. At the same time, our choice to start from local thresholds, which are computed
as a percentile of the time series in the grid point, allowed us to maintain a definition of "extreme" relative to the local
ecosystem properties.
For a biogeochemical variable evolving over two-dimensional space, we proposed a set of indexes for EEWs to describe their
initiation, duration, total covered area and (spatio-temporal) uniformity, as well as their (mean) severity and anomaly, as
measures of overall intensity and inter-annual variability, respectively.
In the specific application to surface chlorophyll in the open sea areas of the Mediterranean, we characterised the top 1% values
of chlorophyll distribution as EEWs that potentially influence ecosystem functions. Cluster analysis conducted on the indexes
associated with the covered areas allowed us to identify four main regimes. We recognised the occurrence of chlorophyll
EEWs with high mean severity and high inter-annual variability in the north-western Mediterranean Sea; chlorophyll EEWs
with high inter-annual variability (associated with intermediate intensity) occurred in the northern Ionian Sea; regular and
spatially well-localised chlorophyll EEWs occurred in the Alboran and south-western Mediterranean Sea, south Adriatic Sea
and the Rhodes Gyre; and weak and diffuse chlorophyll EEWs occurred in the south-eastern Mediterranean Sea.
We did not observe significant trends (i.e., annual variations higher than 1%) of the mean severity of chlorophyll EEWs across
the Mediterranean basin, whereas some trends were found for other indexes.
Comparison of the results with available data and previous studies supports the reliability of the method, which could be
promisingly applied to other ecosystem variables. However, sensitivity analyses are recommended to select suitable thresholds
to highlight the typology of the extreme events under consideration.

## Appendix A: extreme events of surface chlorophyll in Mediterranean Sea regions

Figures A.1-A.3 show time series and distribution of surface chlorophyll in selected sites of the Mediterranean Sea (Fig. 3). The extreme events are identified by the surface chlorophyll values higher than 99th percentile threshold computed in the sites, i.e., by the upper tail (top 1%) of the corresponding distribution. The selected sites show different 99th percentile thresholds, as well as different values and temporal occurrence of the local extreme events. Despite in each site the highest annual values of surface chlorophyll well exceed the maximum values of the climatological seasonal cycles, they do not necessarily correspond to extreme events. Extreme events are in fact identified in some years only.

## Appendix B: ecosystem dynamics of the chlorophyll EEW in the NWM in 2005

Fig. B.1 displays area $A$ (Sect. 2.1.2) covered by the EEW already shown in Fig. 4, and three points, which are internal, peripheral and external to the area (i.e., points A, B and C, respectively). Figures B.2-B.4 display the time series of physical and biogeochemical modelled variables (i.e., heat flux, mixed layer depth, potential temperature, nitrate and chlorophyll concentrations) at the three points. In each panel of the three composed figures, the data from January-April 2005 are compared with the corresponding climatological means computed from 1994-2012.

At internal point A, 30-40 days preceding the EEW onset were characterised by strong heat losses up to 1000 W/m$^2$ (top panel of Fig. B.2) due to the wind field (not shown). This condition led to a strong deep convection that mixed the entire water column down to the sea bottom (second panel of the same figure). Surface and subsurface nitrate, whose concentration at the beginning of the year was already above the climatological values, was further enhanced during the mixing (fourth panel). As soon as stratification was quickly established (second panel) and the surface temperature rose (third panel), an abrupt rise in surface chlorophyll occurred (bottom panel). The surface chlorophyll in January, February and the first half of March exhibited values much lower than the climatological values due to the strong convective phase; in the third week of March, chlorophyll increased by a factor of almost 800% in 4 days. Full consumption of surface nitrate can be observed on the same days (fourth panel). A subsequent weaker mixing phase (in half of April) replenished the surface layers with a relatively low amount of nitrate (yet above their climatological values), triggering two weak episodes of increasing chlorophyll. Overall, the features described here are in agreement with the characterisation of the chlorophyll blooms in the NWM area and, in particular, of the 2005 event (Barale et al., 2008; Estrada et al., 2014; Mayot et al., 2016).

The interpretation of the results referring to peripheral point B belonging to the EEW area (Fig. B.3) is similar to the previous considerations about internal point A but in the presence of less intense vertical mixing.

On the other hand, at point C (Fig. B.4), which is external to the EEW area, an evident stratification of the water column below 30 m depth was maintained for throughout the winter months (January-February-March), despite the cooling of the surface layers. The nitrate content in the surface and subsurface layers was much lower than that in the deeper layers, and only a small increase in the surface chlorophyll developed during the duration of the EEW.

Therefore, the strong deep convection (related to the inter-annual variability of the local vertical mixing) appears to be the key
factor for the exceptionality of this EEW. It is worth noting that our method identified the spatio-temporal region covered by
the EEW (i.e., points A and B) and was shown to effectively include only the relevant information by filtering out other regions
characterised by different dynamics (such as point C).

**Author contribution**

VDB, GC and SS conceived and developed the methodology applied in this work. VDB conducted the formal analysis and
prepared the manuscript with contributions from all co-authors. CS supervised the research activity.

**Competing interest**

The authors declare that they have no conflicts of interest.

**Acknowledgements**

This work was partially sponsored by OGS and CINECA under the HPC-TRES programme award number 2016-04, and a
first formulation of the method was included in Di Biagio (2017). We acknowledge the CINECA award under the ISCRA
initiative for providing high-performance computing resources and support (IscraC codes: HP10CPBEJO, HP10C5FMRK,
HP10COFIJC). The comparison with ESA-CCI data was conducted using E.U. Copernicus Marine Service Information
(http://marine.copernicus.eu/).

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

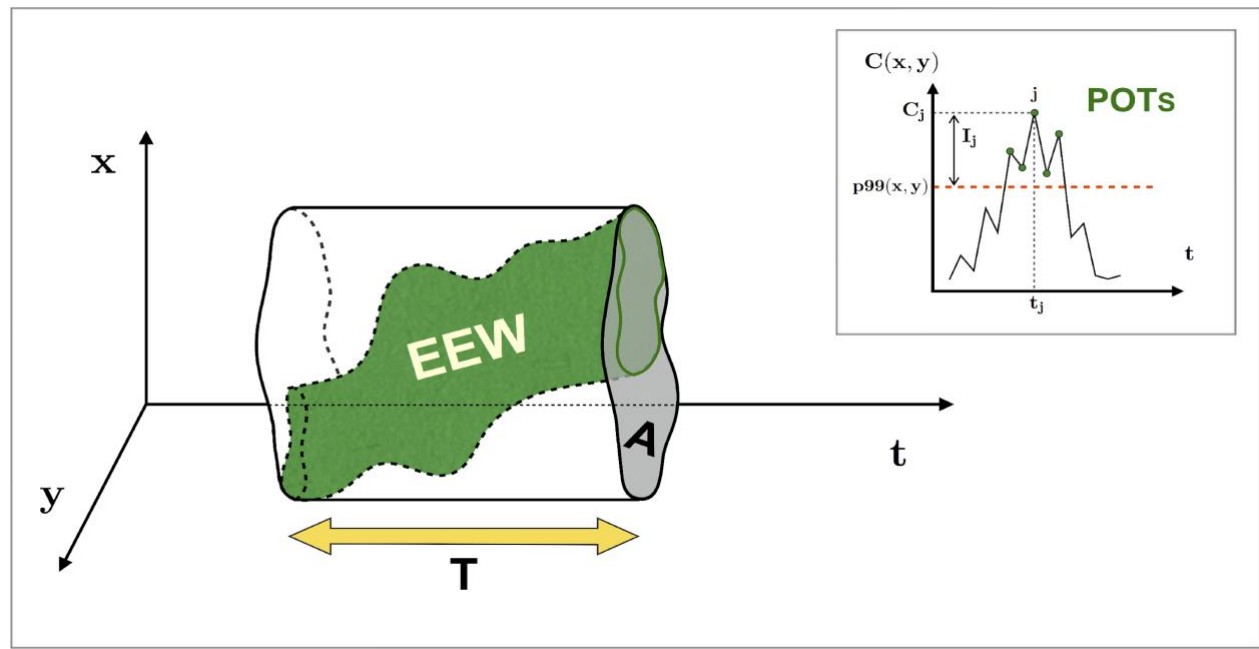

**Figure 1: Conceptual diagram of the spatio-temporal indexes of an EEW (green shape) as a region where the POTs are connected**
**in time and space (top right box). The area and duration of the EEW are indicated by $A$ and $T$, respectively. The uniformity $U$ is the**
**percentage of the spatio-temporal region occupied by the EEW with respect to the total spatio-temporal region of the prism with $A$**
**as the base and $T$ as the height. In the top right box, the POTs (green circles) at the grid point $(x,y)$ are identified by the daily values**
**of the variable above the 99th percentile threshold (orange dashed line). The value of the ecosystem variable $C_j$ and the intensity $I_j$**
**related to the POT index $j$ are also shown.**

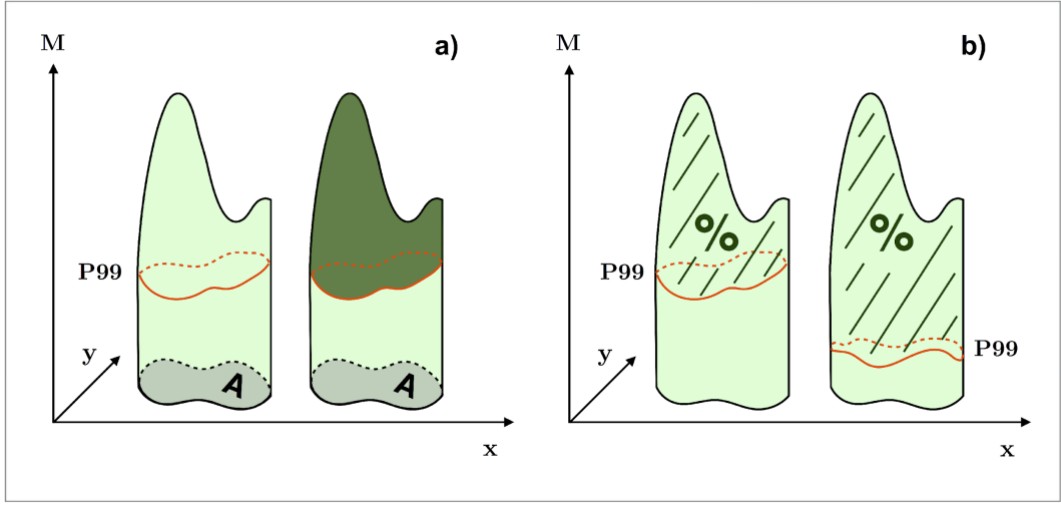

**Figure 2: Conceptual diagrams of the strength indexes: severity and excess (a) and anomaly (b). Fig. 2a): The severity (represented**
**by the shaded green volume, on the left) is the sum of all the $M$ values over each $(x,y)$ grid point belonging to base $A$. The excess (dark**
**green portion, on the right) is the part of this volume that is above the locus of points of the 99th percentile threshold, i.e., $P99$,**
**delimited by the orange contour. Fig. 2b): The anomaly, as the percentage of the excess with respect to the severity, is compared**
**between two cases, which refers to two EEWs with the same severity but with different P99 loci of points.**

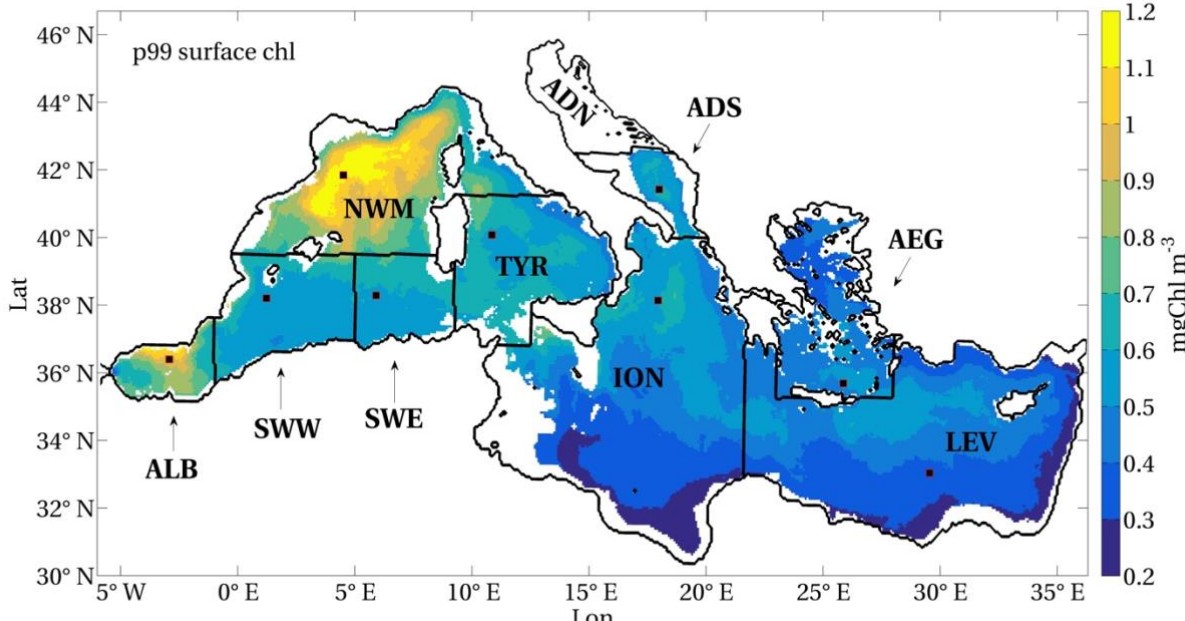

**Figure 3: Model-derived 99th percentile thresholds of the surface chlorophyll in the Mediterranean open sea domain (1994-2012).**
**Isolated grid points with depths higher than 200 m in the Northern Adriatic Sea and the Gulf of Corinth (Greek inlet) are masked.**
**Mediterranean regions delimited by black contours (as in Lazzari et al., 2012) are indicated in the text for simplicity by the**
**corresponding abbreviations (ALB = Alboran Sea, SWW = western side of the south-western Mediterranean Sea, SWE = eastern**
**side of the south-western Mediterranean Sea, NWM = north-western Mediterranean Sea, TYR = Tyrrhenian Sea, ION = Ionian Sea,**
**LEV = Levantine Sea, ADN= northern Adriatic Sea, ADS = southern Adriatic Sea and AEG = Aegean Sea). Black squares indicate**
**sites chosen as examples to display extreme events of surface chlorophyll in the Mediterranean regions (Appendix A).**

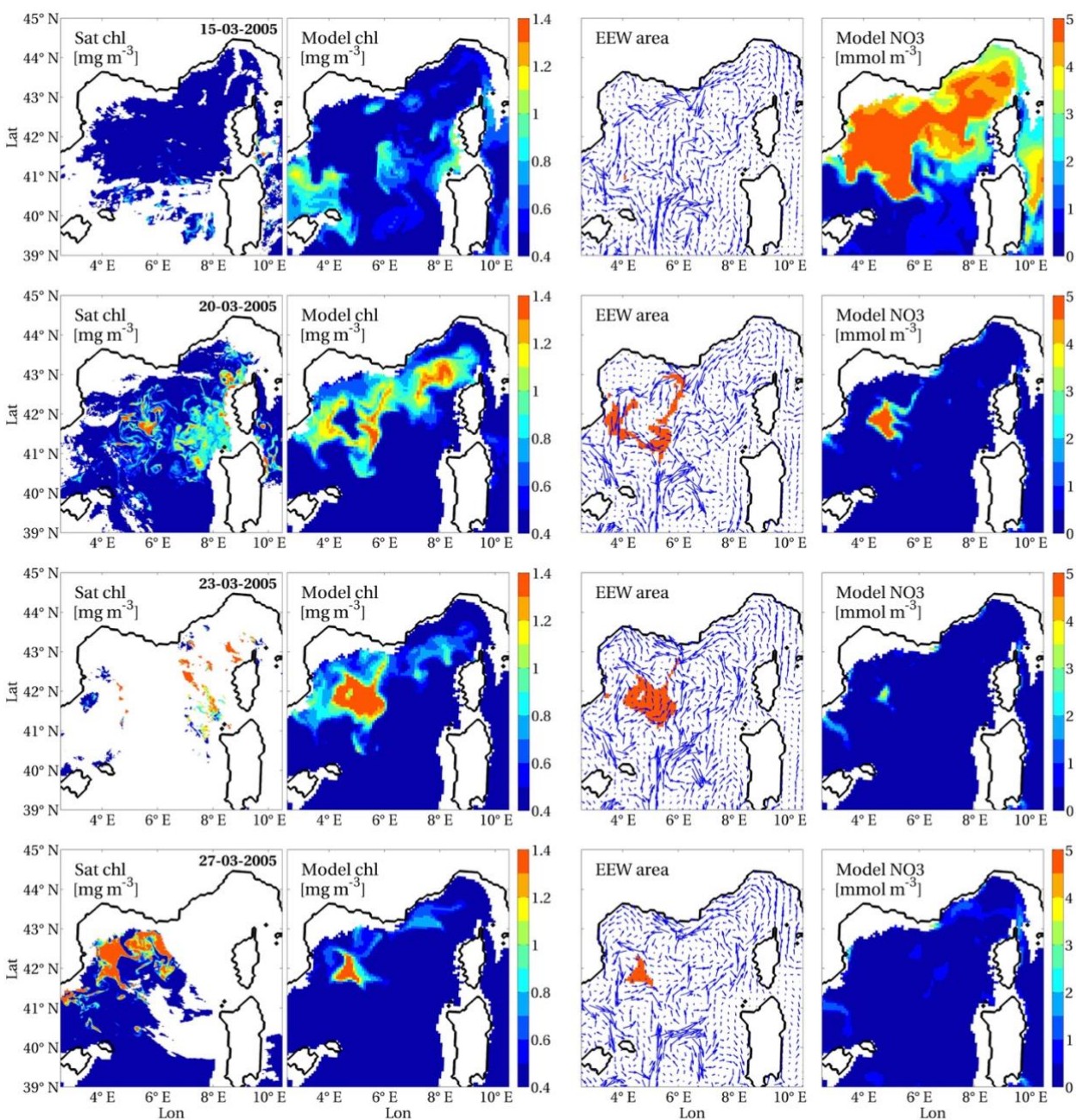

**Figure 4: Surface chlorophyll from satellite (Volpe et al., 2019, first column) and model-derived estimates: surface chlorophyll (second column), daily portion of area *A* of the EEW superimposed on the surface velocity field (third column) and surface nitrate (fourth column), for single days of development of the EEW that occurred in the western Mediterranean Sea from the 15th to 31st March 2005 (indicated in the first column). Surface chlorophyll and nitrate were averaged in the first 10 m of depth and are shown only in the open sea, whereas the horizontal velocity field (scaled by a factor of 1.5) refers to the depth of 5 m.**

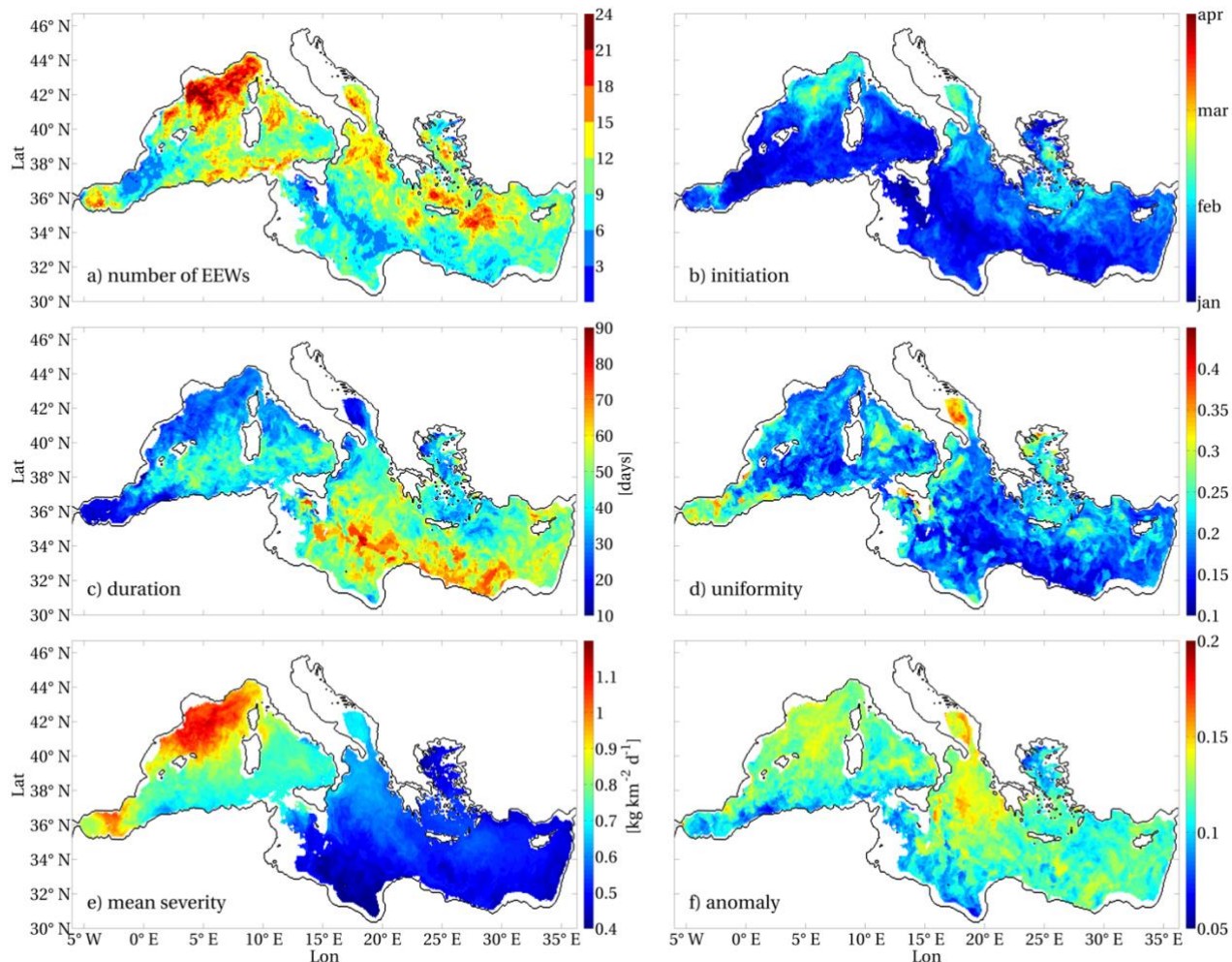

**Figure 5: Number of surface chlorophyll EEWs that occurred in Mediterranean Sea in 1994-2012 (a) and means of the indexes**
**referring to the EEWs: initiation (b), duration (c), uniformity (d), mean severity (e), anomaly (f). The values of the indexes referring**
**to an EEW were associated with all the points belonging to covered area** *A* **(Sect. 2.1.2).**

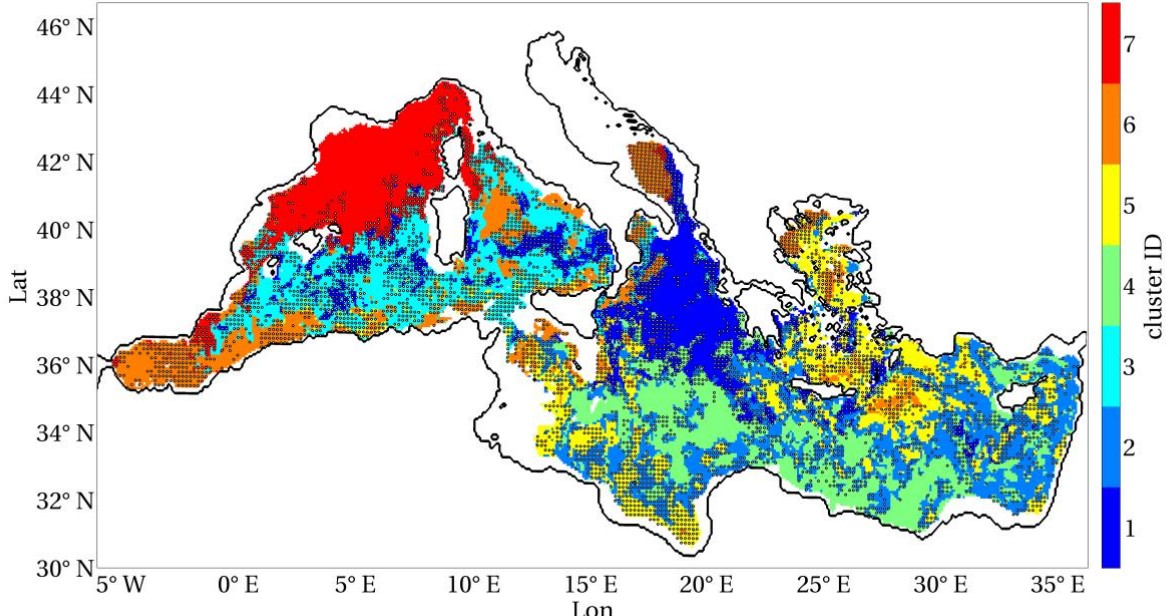

**Figure 6: Fuzzy clusters with maximum membership identified from the duration, mean severity, uniformity and anomaly maps**
**(shown in Fig. 5), with black points indicating a confusion index higher than 0.7.**

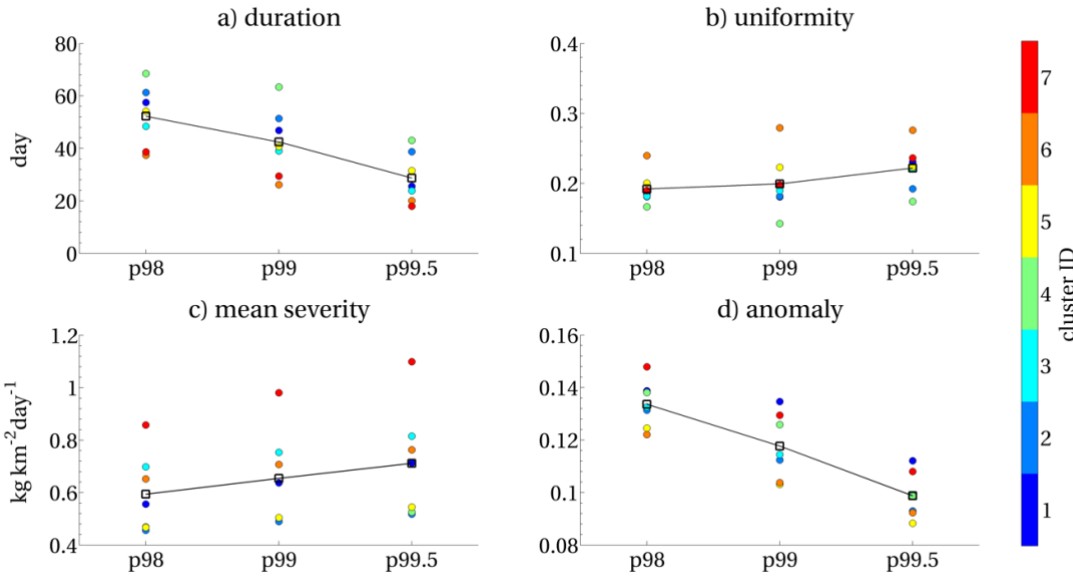

**Figure 7: Means of duration, uniformity, (mean) severity and anomaly computed for the surface chlorophyll EEWs obtained from**
**different local thresholds (Sect. 2.1.1), with the 99th percentile (p99) as reference used in the present study. Coloured dots represent**
**the mean values of the indexes computed in the corresponding clusters (with the same colour legend of Fig. 6). The total means**
**(black squares) are computed by averaging the means of the seven clusters.**

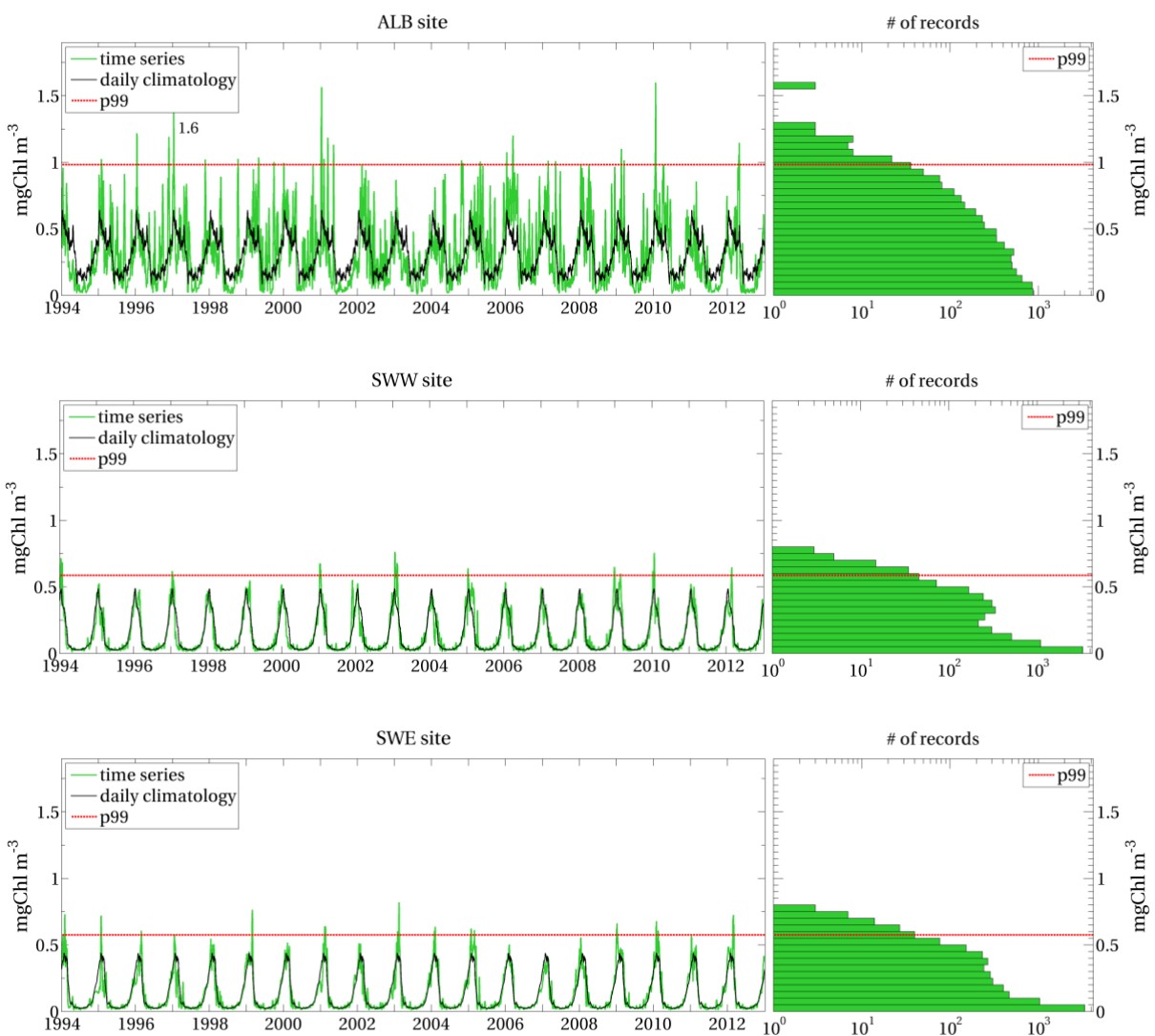

**Figure A.1: Left panels: time series of model-derived surface chlorophyll (green series) and daily climatology (black series) computed**
**in ALB, SWW and SWE sites indicated in Fig. 3; right panels: model-derived surface chlorophyll distribution in the corresponding**
**sites, with x axis in logarithmic scale. Red dashed lines represent the local 99th percentile thresholds in all panels.**

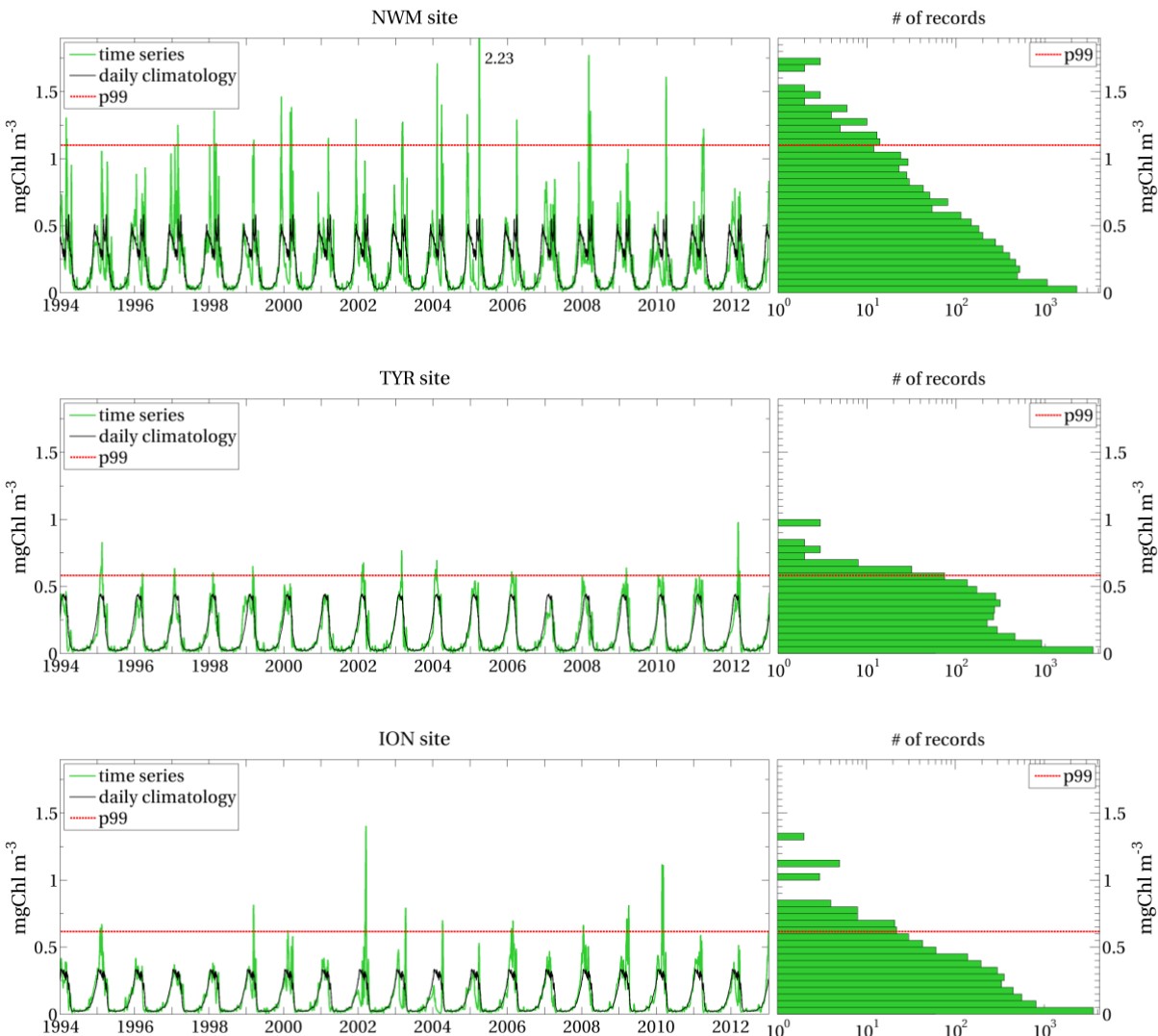

**Figure A.2: As in Fig. A.1, but referred to NWM, TYR and ION sites, respectively.**

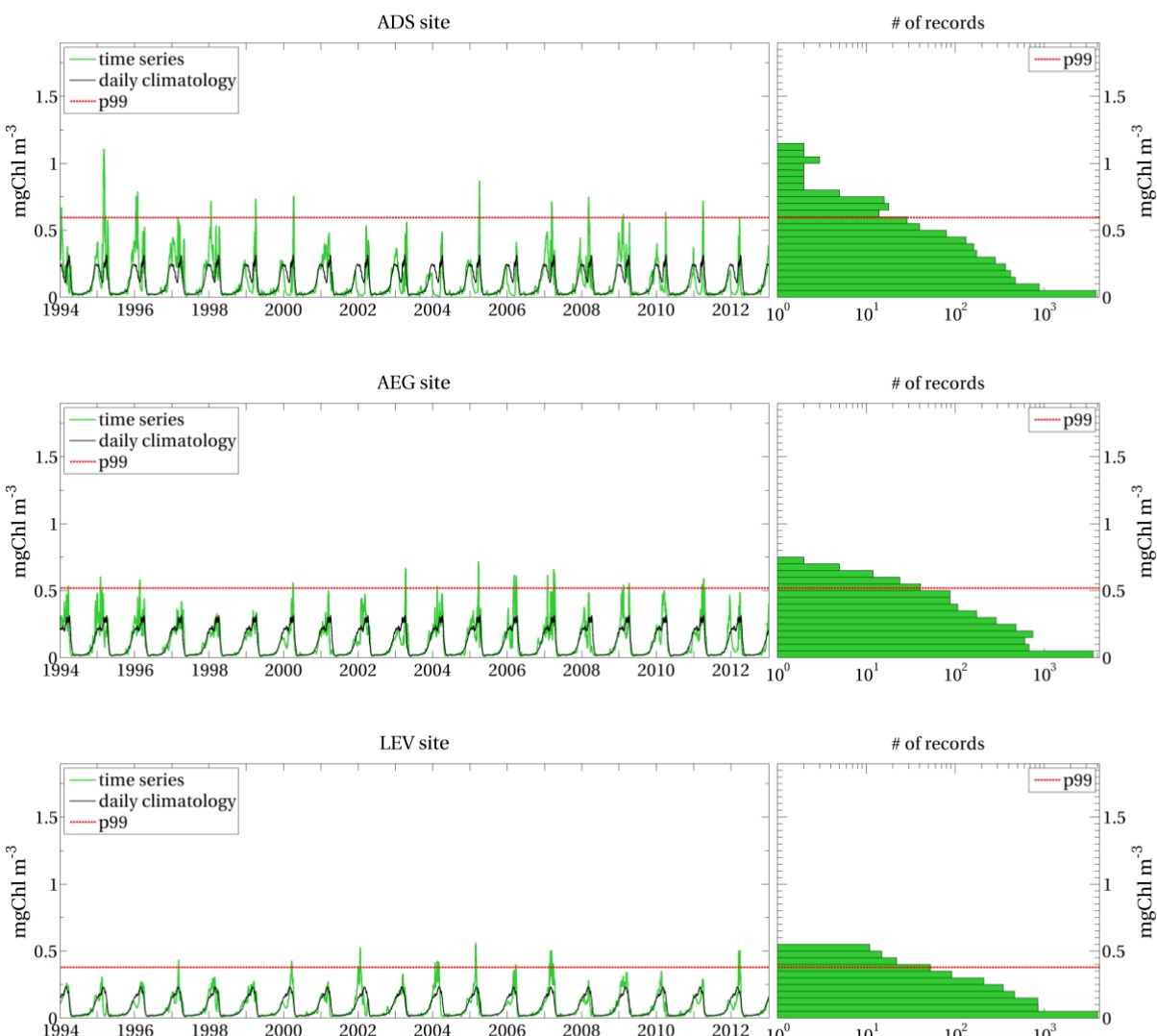

**Figure A.3: As in Fig. A.1, but referred to ADS, AEG and LEV sites, respectively.**

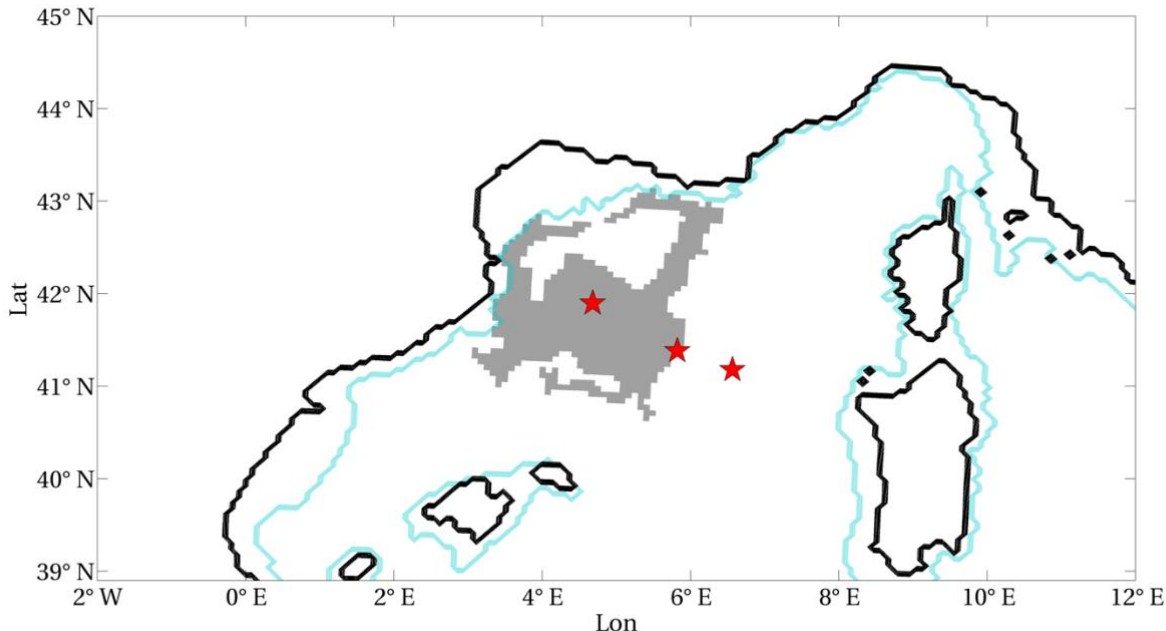

**Figure B.1: Area *A* of the surface chlorophyll EEW shown in Fig. 4 (grey patch). Red stars indicate grid points in internal (A),**
**peripheral (B) and external (C) positions with respect to the EEW. Blue contours delimit the open sea area (i.e., depths greater than**
**200 m).**

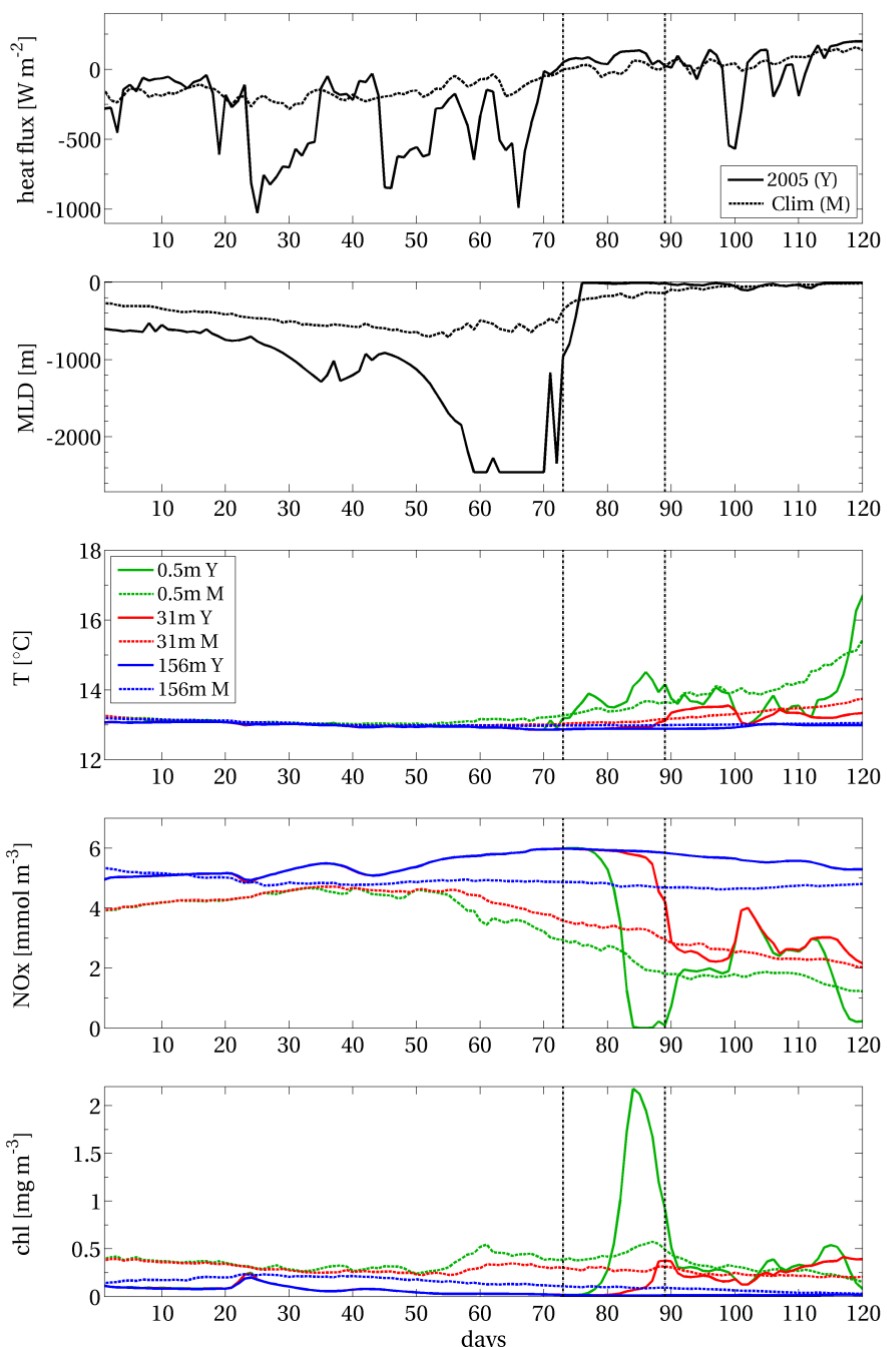

**Figure B.2: Net surface heat flux, with negative sign for ocean cooling, mixed layer depth and potential temperature and**
**concentrations of nitrate and chlorophyll at different depths (see the legend in the central panel), computed at the internal point A**
**of the EEW in NWM (Fig. B.1). Acronyms: Y = year (2005), M = (climatological) mean. Days are computed from the 1st Jan 2005.**
**The vertical dashed lines delimit the duration of the EEW.**

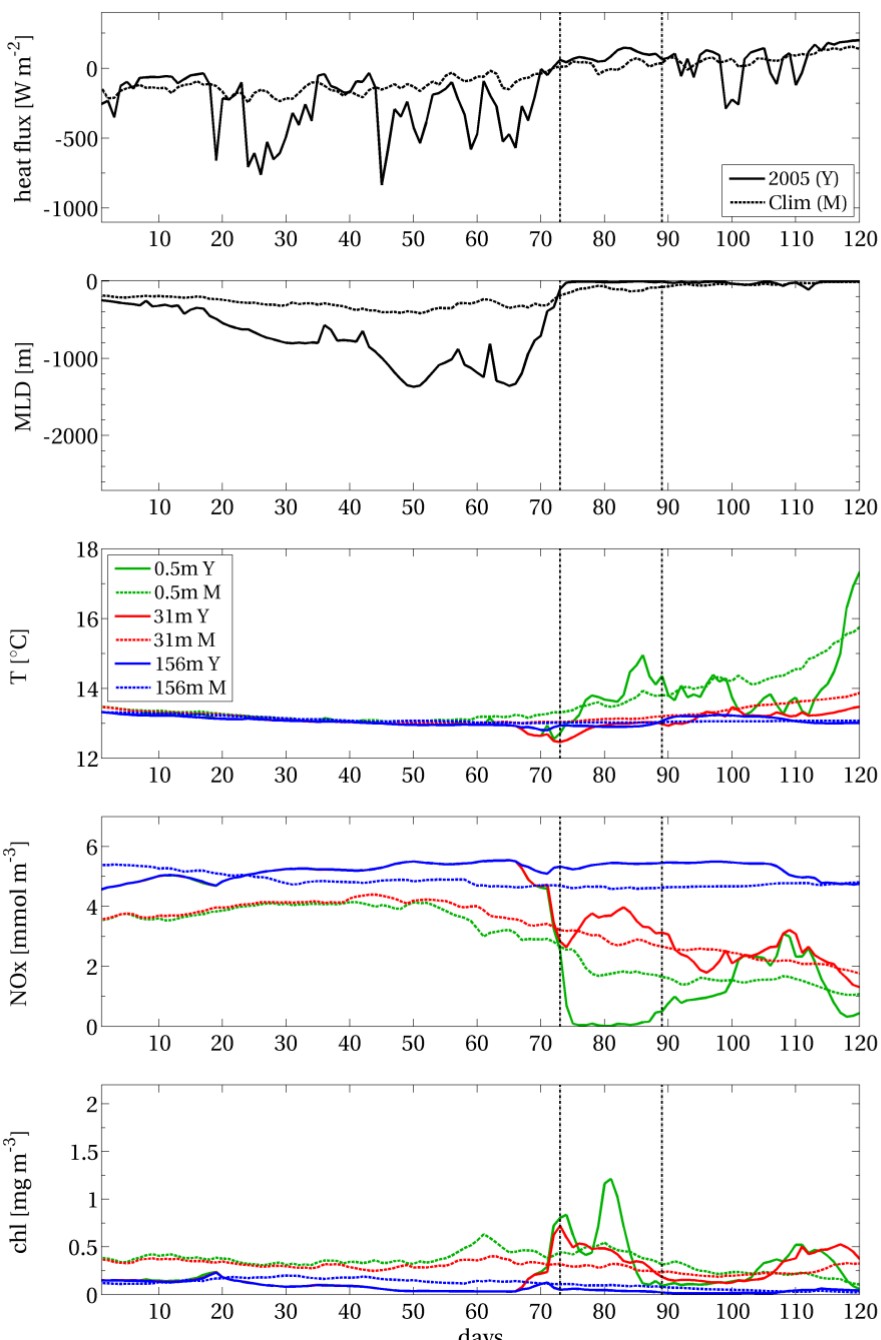

**Figure B.3: As in Fig. B.2, but in the peripheral point B of the EEW area in Fig. B.1.**

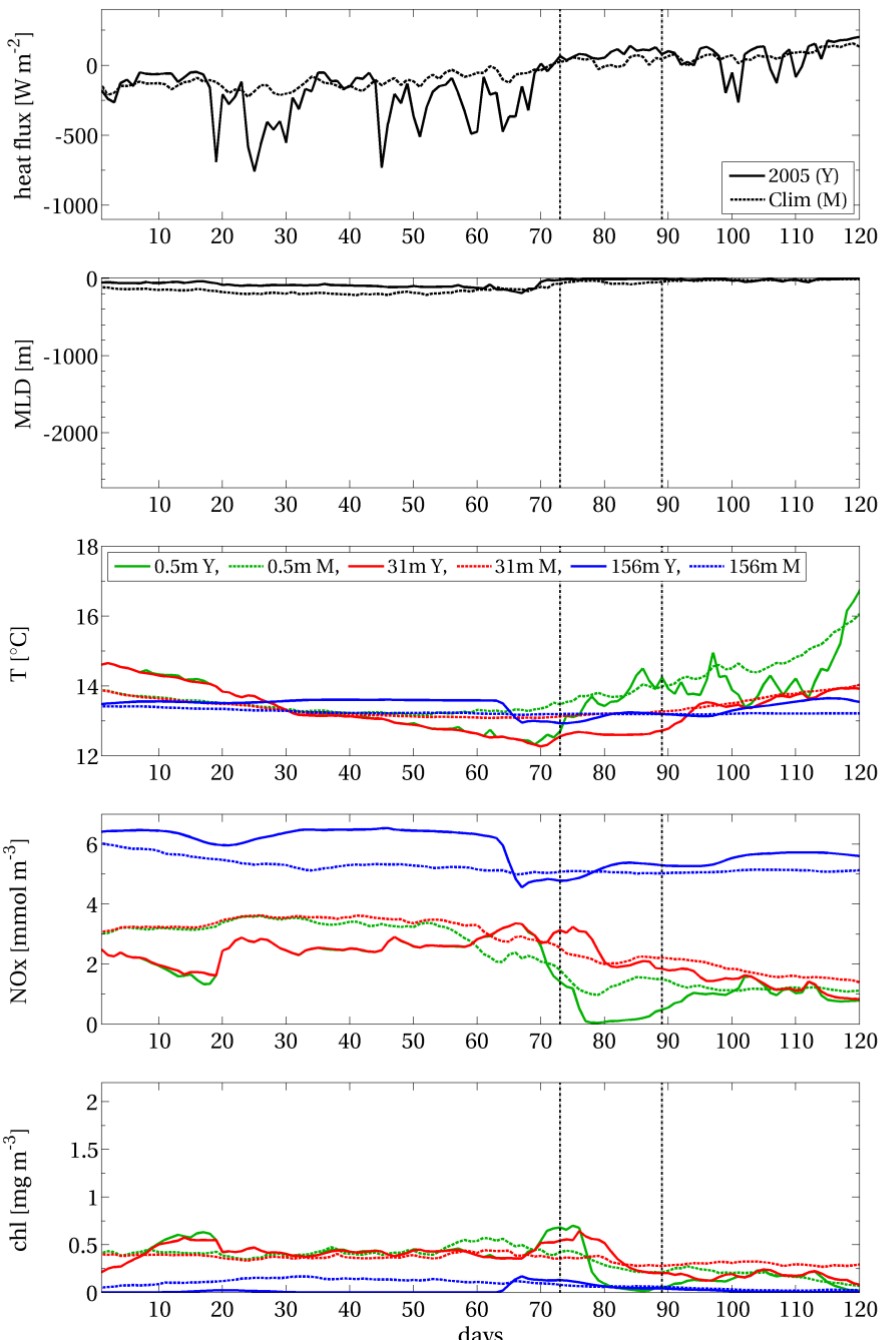

**Figure B.4: As in Fig. B.2, but in the point C external to the EEW area in Fig. B.1.**

| Spatio-temporal | Initiation | 15th March 2005 |
|---|---|---|
| | Area $A$ [$km^2$] | $33.1 \times 10^3$ |
| | Duration $T$ [$day$] | 17 |
| | Width $W$ [$km^2 \times day$] | $1.06 \times 10^5$ |
| | Uniformity $U$ | 0.189 |
| Strength | Severity $S$ [$kg$] | $1.479 \times 10^5$ |
| | Excess $E$ [$kg$] | $3.032 \times 10^4$ |
| | Mean Severity $< S >$ [$kg\ km^{-2}day^{-1}$] | 1.389 |
| | Anomaly $AN$ | 0.205 |

**Table 1: Metrics, grouped by spatio-temporal and strength indexes (defined in 2.1.2), for the EEW in Fig. 4.**

| #Cluster | Duration $T$ [$day$] | Uniformity $U$ | Mean severity $<S>$ [$kg\ km^{-2}\ day^{-1}$] | Anomaly $AN$ |
|---|---|---|---|---|
| 1 | 47±6 | 0.181±0.027 | 0.637±0.087 | 0.135±0.008 |
| 2 | 51±4 | 0.182±0.017 | 0.490±0.042 | 0.112±0.008 |
| 3 | 39±5 | 0.190±0.026 | 0.754±0.051 | 0.115±0.008 |
| 4 | 63±6 | 0.142±0.015 | 0.505±0.040 | 0.126±0.009 |
| 5 | 41±5 | 0.223±0.020 | 0.505± 0.067 | 0.103±0.011 |
| 6 | 26±7 | 0.280±0.040 | 0.707±0.135 | 0.104±0.017 |
| 7 | 29±5 | 0.198±0.030 | 0.981±0.084 | 0.130±0.008 |


**Table 2: Mean and standard deviation of the duration, uniformity, mean severity and anomaly indexes within the seven clusters in**
**Fig. 6. Pale red (blue) refers to an annual increase (decrease) higher than 1% in the 1994-2012 period.**