# Peer review of "Extreme event waves in marine ecosystems: an application to"

_Biogeosciences, 2020_

## Referee Comment (RC1) · Anonymous Referee #1 · 9 Apr 2020

== Overall Comments

In this Study, Valeria Di Baggio et al. use an extreme event identification method to track the late winter-early spring blooms in the Mediterranean sea. Their method enable to identify and follow day by day the bloom propagation, and characterize the event with different indexes.

Although the method is shown to be powerful and useful, I have some questions/concern with the application done here with the Mediterranean surface chlorophyll, as I am not sure what we are looking for, and getting in the end... Are we looking for extremes ? blooms ? strong blooms ? blooms maxima ? maxima of surface chl
maximum ? we are not sure, and the way it is done probably allow all of those. But then... Are blooms considered as extreme events ? Apart from this main and i think important concern, the study is nice and relevant. the way the authors manage to track and characterize these events is shown to be useful, with lots of relevant information, and could be exported for all kind of extreme event study.

I really appreciate this study, but it has to make clear what we are looking at: extreme? or surface chlorophyll maximum ? depending on the answer, the amount of work needed to correct the paper will be different, corresponding to a major review if you want to make it an extreme event analysis; or a minor review if it rather is a surface chl maximum analysis using an extreme event tool (what i think the authors are doing here).

== Extreme, Bloom, or Surface Chlorophyll Maximum ?

Although the method is shown to be powerful and useful, I have some questions/concern with the application done with the Mediterranean surface chlorophyll, as I am not sure what we are looking for extremes ? blooms ? strong blooms ? blooms maxima ? maxima of surface chl maximum ? it seems you see all of those including extremes, like the one you have selected for the example. But then... Are blooms considered as extreme events ? From the definition you give (Page 2, line 35) "a large deviation from a reference state", but i think the reference state should include the annual cycle... if you are looking for extreme. If your targets are extremes of surface Chl, you still could use the 99th percentile threshold, and only keep those going above the local annual cycle + STD (or 1.5 * STD) for example, or instead of a 2D threshold make it 3D, including an annual cycle, that could also show extreme in summertime (maybe due to dust events for example),... The choice of a 2D 99th percentile on the whole period is somehow too broad if you look for extremes, and of course you will get completely different results in the different area of the Mediterranean sea. In the North-Western part with strong and spiky blooms,you will overshoot the threshold at least once a year, because of this spiky bloom configuration (but that's not extreme... it

is the every year bloom), whereas in the oligotrophic region, where the chl phenology is smooth, a relatively stronger event can cover the whole 99th percentile.

So, if you were looking for extreme, you should change that, adapt the threshold which seems to be the key of the method. But i am not even sure you are looking for extreme. The text and the title are confusing. If you are characterizing blooms (or maxima in chl maximum) using an extreme event method, it is great, but present it like this. Please, don't try to oversell it. A title like "Tracking the Mediterranean blooms using Extreme event waves method" or something like that. Of course the current title is more punchy, but personally, when i read it i've imagined dozens of possible things.

Also, make it clear in the text :

-p1 l14: identify the maxima of chlorophyll as exceptionally high and prolonged "blooms"

-p2 l54: This allowed to identify maxima of phytoplankton blooms (Desmit et al., 2018), but also positive anomalies with values too low to be actually considered "bloom""

-p6 l172 : (i.e., exceptionally high and prolonged "blooms", as clarified in Introduction)

-p9 l272 : probably the clearer explanation : "we propose a new method to tackle extreme events in the marine ecosystems on the basin scale. The method is then applied to the surface chlorophyll in Mediterranean open-sea areas to investigate maxima in the winter-spring blooms".

So sometimes it is "exceptionally high and prolonged", some other time it is "maxima in the winter-spring" blooms. The second (which includes the first) sounds more accurate, but read both is confusing. please make it clearer.

What struggles me is the lack of definition for bloom and for bloom maxima, Or at least what you consider "blooms" and "bloom maxima" in this study. what gives me the impression of not being sure of what we are looking for, and results with places where a bloom maxima appears every year, and other places where it happens once or twice

in 18 years and last 90 days. In the oligotrophic region, where there is no blooms, an EEW is found by construction (as said p6,l181 : "Considering the temporal extension of the simulation (approximately equal to 7000 days), the number of POTs in each grid point is by construction equal to 70.") the long EEW might well be an eddy with higher surface chl concentration inside. It cannot be considered a bloom.

A solution could be to:

– Stop talking about blooms for the whole Mediterranean sea. It would make more sense if you were talking of "(...) investigate maxima in the winter-spring surface chlorophyll maximum". That would be more correct, the maxima being not necessarily extreme, and not saying the word "bloom" don't mislead the attention on something specific that does not occur everywhere in the Mediterranean sea.

– And stop talking about extremes everywhere. The method you use is a method that is first made to find extreme events, but the way you use it, you don't only find extremes. an extreme event that comes back at least once a year is not an extreme, it is part of the normal annual cycle.

– Something else that could help to better visualise how extreme the EEW are. You could try to plot the surface Chl annual cycle (with STD in dashed line) for each Mediterranean regions (Fig 3), with the averaged 99th percentile threshold represented on top. that way we can appreciate how "extreme" an EEW is for each area (Maybe you want to adapt the area so it looks more like the fig 6 ? might be more relevant).

Unless you want to talk about extremes and only extremes. Then you have to adapt the threshold by taking into account the surface chlorophyll annual cycle as suggested above.

Apart from this (important) semantic question, the method is nice and prove to be able to identify, characterize and track the EEW beautifully.

– Also, talking about extremes, i wanted to rise a question, just for discussion. I understand the choice of surface Chl maxima is mainly to test the method and show how it works. But thinking about Mediterranean sea, climate change and extreme events, i wonder if tracking maxima of surface chlorophyll maximum is what i would do. I don't think we can get hypoxia or eutrophication with 12th degree model, this is rather a coastal and river mouth problem. We know that a climate impact could be to lower the deep water formation and hence the bloom. We could use your method (adapting the threshold, considering the Annual cycle) to track years with little or no bloom, and understand why, and see the trends. Or in summertime if your model include dust deposition on high frequency, see if the model shows EEWs linked to dust deposition events,... There is lots of other application of your methods that could make lots of sense (Lots of nice study in perspective).

== Text remarks

– I think there are few places where the English could be corrected, but not being a native English myself, i am not the right person to do that. Maybe you could ask a native English around you to double check your manuscript.

– from p5.l134 and all units following : double check the units the -2 and -1 should be up, if you write with latex, you should write kg km^{-2} day^{-1}

– p6 l73 : " chlorophyll as a proxy for the phytoplankton biomass" Surface chlorophyll is representative of the surface biomass (probably why one of your idea in the discussion is to check the event in 3D)

– p6 l82 to 85: "Mapping the 99th percentile threshold values computed at each grid point on the whole basin (Fig. 3), it can be noticed that grid points that are near in space exhibit small differences in their threshold values and also that different patterns are recognisable in the basin. Hereafter, we use the abbreviations indicated in Fig. 3 to refer to different Mediterranean regions" – So the 99th percentile is fixed in time. This means that you compare toward an ∼annual 99th percentile threshold of Chl. basically you will only have EEW during the bloom period. A summer with exceptional

summertime Chl will not appear with this method as it will never exceed bloom period values. can't you do a time varying 99th percentile threshold to be able to see non-bloom period EEW ? otherwise you will probably miss the most interesting events... probably needs a longer run to get enough data to keep it statistically feasible.

– p7 l191 : " The model-derived chlorophyll patterns (Fig. 4, second column) are in good agreement with the remote sensing data (first column) in the same temporal interval of the EEW" – Hard to tell, seems the sat Chl has a more extended bloom than the model, and starts slightly later (and probably ends later as well). But both model and sat presents an EEW on the same period, what is already a nice model performance ! And you have a nice bloom in the Ligurian sub-basin, that's impressive! Talking about e Ligurian bloom, it does not appear in the EEW area. it is considered as a separated EEW ?

– p8 l235 : " are around half of the ones of ALB or NWM." needs to be rephrase.

– p8 l239 : "with a similar chlorophyll EEWs phenomenology."

– p10 l288 : "pointed out the heterogeneity of the blooms intensity in the Mediterranean Sea" - back to my main comment, you don't see blooms everywhere...

– p10 l310 : Furthermore ?

– p10 l310 : you could have shown the "spatio-temporal persistence", it looks like a nice index. Why not show it ?

– p12 l360 to 374 : Good idea!

– p12 l375 : "A critical parameter of our method is the choice of the local percentile threshold" - I agree looks like one of the key of the method. but why this choice of a simple percentile threshold, and not include the local annual cycle ( maybe + a*STD ) in the threshold (As i mentioned above depending what you want to analyse, it can be justified, it can be the right choice) ?

– p13 l 197 : "Of the clusters with the highest content of all the indexes has been generally maintained both in case of higher and lower thresholds" - rephrasing : of the clusters with the highest index values,...,

– p13 l396 : "A key issue" - not issue, it is one of the strength of this method, not issue i think, and from all what you could do in your study because of that.

– p13 l400 : "The time series in the grid point" - rephrase : Each grid point's time serie

– p13 l400 : " allowed to maintain a definition of "extreme" relative to the local ecosystem properties." This i do not agree. in some places like the most oligotrophic regions, you probably find extremes, but in the bloom regions, it is not.

– P23 - Fig 4 – You talk about the MLD in the text, but you don't show it on the plot. Of course, we can guess the Mixed layer is very deep where the NO3 is high and Chl low, but, it might be good to add iso-contour with depth values on the Chl plot for example. That would help both the writer and the reader. – very nice bloom in the Legurian sub-basin! You must have a very high res atm model with high freq coupling. you should add these details in the model description. It help understand the results.

– P24 - Fig 5a – Difficult to interpret.... and the color-scale does not help. how many EEW occur per year ? how many on the hole period ? I don't understand what you mean here.

– P27 - Fig A.2, A.3, A.4 – I cannot do the difference between the climatological line end the 2005 one. Please, try with different dashed or doted line to find one that we really can see.

---

## Author Comment (AC1) · 27 Apr 2020

We thank Reviewer 1 for his/her insightful comments, which we will thoroughly take into account in our full revision. Here we would just like to briefly clarify some key aspects regarding the focus of the paper.

This study aims to propose a methodology for the analysis of "extreme event waves", defined as a set of "extreme events" which are contiguous in space and time.

To do this, we started from the statistical definition of "extreme event" in each and every point (x,y): at each point, an extreme event is an event that occurs very rarely (in our

example less than 1% of the number of observations) and it is statistically defined as a value over a predefined percentile threshold (in our example arbitrarily chosen to 99th percentile), which is computed with reference to observations (time series) recorded at that specific point (i.e., the threshold is site specific, $p99(x,y)$). This definition can be applied to any time series, and also to deseasonalized time series, depending on the scientific question. If the methodology is applied to a variable which is driven by a strong seasonal signal, the extreme events might result more or less "regularly" distributed over the years, however they still are "rare events" (the top 1%).

Then, we introduced the concept of "wave" of extreme events, as a set of those extreme events which are connected in space and time. The paper focuses on the identification of these "extreme events waves" (EEWs) and on their characterization, by means of some novel indexes (e.g. duration, severity, anomaly). In particular, our method is able to distinguish between EEWs which occur regularly/irregularly over the years, by means of low/high values of our anomaly index, respectively. Moreover, our method distinguishes between EEWs which occur with high/low absolute values of the chosen variable, by means of high/low values of our severity index, respectively.

We selected the chlorophyll as the variable of application of the method. The time series of chlorophyll show impulsive dynamics, modulated in time (mainly by the seasonal cycle) and dependent on the location (i.e., geographical point). We observed that extreme events computed on the time series of chlorophyll occur quite regularly over the years in many points, but not everywhere: in northern Ionian Sea, for example, they occur only in some years. One of our indexes (i.e., anomaly) has been designed to track this feature in the EEWs. Indeed, the strong inter-annual variability in northern Ionian Sea is well detected by EEWs with high anomaly values. Moreover, the EEWs of chlorophyll are associated both with "blooms" (e.g. in the Gulf of Lion) and with chlorophyll values which are too low to be properly considered as "blooms" (e.g. in southern Levantine Sea). Our method is able to distinguish between these two cases, by means of high and low values of the severity index, respectively. However, we are

aware that our use of the term "bloom" was quite confusing and we will avoid it in the revised manuscript.

On the other hand, we will carefully revise the use of the "extreme" term in the text, to further clarify our terminology, by highlighting the points above.

―――――――――――――――

---

## Referee Comment (RC2) · Anonymous Referee #2 · 5 May 2020

The manuscript "Characterisation of extreme events waves in marine ecosystems: the case of Mediterranean Sea" describes a new method to characterize extreme events bases upon simulated chlorophyll concentrations for the period 1994-2012. Using a cluster analysis applied to a set of indices that define the occurrence of extreme events waves, different ecosystem regimes were defined. In my opinion, the manuscript is very interesting and deserves publication in Biogeosciences after minor to moderate revisions. In detail: 1) The language should be checked by a native English speaking person. There are several typos that should be removed. For instance in line 270 there is a reference to Fig. 9 that does not exist. 2) The method uses surfaces chlorophyll concentrations averaged over the uppermost 10 m and does not consider vertical profiles. Please discuss why subsurface blooms do not play a role. 3) It is discussed but I am still worried about the stability of the identified regimes using different thresholds. You showed that the clustering of the mean values of the indices do not change much. However, are there changes in the spatial distribution of the regimes shown by Figure 6? 4) In Section 4 and the Appendix, ecosystem dynamics characterizing some of the regimes are discussed. However, some clusters lack any dynamical explanation and might be rather artificial. It would increase the scientific value of the manuscript if you could discuss these clusters more in detail as you have done it for NWM in the Appendix. 5) In the abstract you mentioned that "There is a growing interest about events that can affect ecosystem functions and services in a changing climate". However, is the method suitable for following the temporal shifts in the regimes without any discontinuities? I suggest that you split the period in half and that you use the cluster analysis for both periods. By this, the impact of trends in some of the indices on the spatial distribution of the regimes might be investigated and compared with observed changes.

---

## Author Comment (AC2) · 26 May 2020

We sincerely thank Reviewer#1 for his/her comments, which gave us the opportunity to clarify some key points of our paper.
We indicate our reply in blue colour and some corrections we propose to implement on the text of the manuscript in italic red. The full version of our reply, if accepted, will include all the corrections declared here.

**== Overall Comments**

In this Study, Valeria Di Baggio et al. use an extreme event identification method to track the late winter-early spring blooms in the Mediterranean sea. Their method enable to identify and follow day by day the bloom propagation, and characterize the event with different indexes.
Although the method is shown to be powerful and useful, I have some questions/concern with the application done here with the Mediterranean surface chlorophyll, as I am not sure what we are looking for, and getting in the end... Are we looking for extremes ? blooms ? strong blooms ? blooms maxima ? maxima of surface chl maximum ? we are not sure, and the way it is done probably allow all of those. But then... Are blooms considered as extreme events ?

This study aims to propose a methodology for the analysis of "extreme event waves" (EEWs), defined as a set of "extreme events" which are contiguous in space and time.
We defined statistically an "extreme event" in each point $(x,y)$ as a value of the variable which is over a given threshold. In our case, the threshold is set as the 99th percentile of the time series recorded at that specific point (i.e., the threshold is site specific: $p99 = p99(x,y)$).
Extreme events are thus represented by the maxima of the variable distribution in that point (i.e., represented by the upper tail of the distribution). They are "rare" events (i.e., their number is equal to 1% of the total records; right panels in Fig. R.1.1 and R1.2), selected independently from their distribution over the years. In fact, the temporal regularity over the years (i.e., the inter-annual variability) is one of the features which, in retrospect, can be quantified in the analysis of the "extreme events waves" by means of our "anomaly" index.

In our specific application focused to surface chlorophyll, the "extreme events" are strictly defined as the maxima of the local distribution of surface chlorophyll, corresponding to the highest values recorded in the point. These maxima of the distribution can be distributed quite regularly over the years and thus correspond to annual maxima (i.e., in the seasonal cycle), or they can spread among a few years, as in case of the northern Ionian Sea (as shown in Fig. R1.1, below). In the latter case, such high inter-annual variability has been detected in the EEWs covering that area, by means of high values of the anomaly index.
On the other hand, the maxima of the chlorophyll distribution can correspond to blooms (e.g., in the Gulf of Lion), but also to chlorophyll values which are too low to be properly considered as "blooms" (e.g., in the southern Levantine Sea). Our method is able to distinguish between these two cases, by means of high and low values of the severity index of the EEWs, respectively.

[Figure]

**Fig. R1.1** Time series of surface chlorophyll in a site belonging to northern Ionian Sea, with daily climatology computed in the site (left) and histogram of frequency of the chlorophyll values recorded in the site (right). In both panels, the horizontal dashed line indicates the 99th percentile threshold computed in the site.

Thus, considering Reviewer#1's objection, we propose a method to analyse the extreme events; based on our analysis, blooms can be extreme events, but not all extreme events can be classified (generally speaking) as blooms.

We are aware that our use of the term "bloom" in the previous version of the manuscript was quite confusing, and we will avoid it in the revised manuscript. On the other hand, we will carefully revise the use of the terms "maxima" and "extreme", to further clarify our terminology, by highlighting the points above.

Apart from this main and i think important concern, the study is nice and relevant. the way the authors manage to track and characterize these events is shown to be useful, with lots of relevant information, and could be exported for all kind of extreme event study.
I really appreciate this study, but it has to make clear what we are looking at: extreme? or surface chlorophyll maximum ? depending on the answer, the amount of work needed to correct the paper will be different, corresponding to a major review if you want to make it an extreme event analysis; or a minor review if it rather is a surface chl maximum analysis using an extreme event tool (what i think the authors are doing here).

As previous clarified, we actually conducted an analysis on the maxima of the surface chlorophyll distribution (i.e., "extreme events", with respect to p99(x,y) of the local chlorophyll distribution) and the subsequent steps of the method (i.e., the identification of the extreme event waves (EEWs), their characterisation and classification) can be defined an "extreme events tool", as Reviewer#1 suggests. Nevertheless, we hold that the validity of the whole method is general, i.e., the same definitions can be applied also to deseasonalized time series (see also the second next point). In that case, the "extreme events" under study would be identified by the values differing the most from the daily climatological mean (where "the most" would be set by the 99th percentile threshold of the distribution of the anomalies). The "extreme events" identified in this latter way do not necessarily correspond to the highest values of the variable recorded in the point and identify local perturbations with respect to the seasonal cycle. The choice to deseasonalize or not the time series depends on the scientific question.
In our analysis, we focused on maxima of surface chlorophyll distribution and limited their characterization and classification to the winter-spring period.
We recognise that the difference between absolute and time-dependent thresholds, as well as the possibility of following the latter approach in the first part of our method, deserves an explicit mention in the manuscript. We propose to add the following sentences in the Discussion section, line 393:
*Finally, it is noteworthy to specify that our method of EEWs identification, characterisation and classification can be applied also to extreme events defined starting from seasonally varying threshold (as e.g. in Hobday et al. 2016). In this case, "extreme events" would correspond to the*

*highest anomalies recorded with respect to the climatological seasonal cycle of the variable, and generally not to the highest values of the variable recorded in the whole time series (as in our case of temporally fixed threshold). Such an application would allow to investigate scientific questions of a different kind, such as chlorophyll anomalies in summer time.*

**== Extreme, Bloom, or Surface Chlorophyll Maximum ?**

Although the method is shown to be powerful and useful, I have some questions/concern with the application done with the Mediterranean surface chlorophyll, as I am not sure what we are looking for extremes ? blooms ? strong blooms ? blooms maxima ? maxima of surface chl maximum ? it seems you see all of those including extremes, like the one you have selected for the example. But then... Are blooms considered as extreme events ?

Please refer to our reply to Overall Comments to clarify the concepts of "extreme", "blooms" and "maxima".

From the definition you give (Page 2, line 35) "a large deviation from a reference state", but i think the reference state should include the annual cycle... if you are looking for extreme. If your targets are extremes of surface Chl, you still could use the 99th percentile threshold, and only keep those going above the local annual cycle + STD (or 1.5 * STD) for example, or instead of a 2D threshold make it 3D, including an annual cycle, that could also show extreme in summertime (maybe due to dust events for example),... The choice of a 2D 99th percentile on the whole period is somehow too broad if you look for extremes, and of course you will get completely different results in the different area of the Mediterranean sea. In the North-Western part with strong and spiky blooms,you will overshoot the threshold at least once a year, because of this spiky bloom configuration (but that's not extreme... it is the every year bloom), whereas in the oligotrophic region, where the chl phenology is smooth, a relatively stronger event can cover the whole 99th percentile. So, if you were looking for extreme, you should change that, adapt the threshold which seems to be the key of the method. But i am not even sure you are looking for extreme.

As we replied also to the Overall Comments, we think that deseasonalizing or not the time series is a choice that depends on the scientific question. In our case, we investigated the highest values of surface chlorophyll defined by the tail of the distribution of chlorophyll in each point. The investigation of "extreme events" of anomalies with respect to the daily climatological mean is a different (and, for sure, interesting) study. It would give us results of a different nature, which would deserve a separate paper. However, our method can be applied also to the deseasonalized time series, since it analyses the spatio-temporal contiguity of "extreme events" and provides a "tool" for their characterisation and classification.
In our analysis, we propose a description of the phenomenology of extreme events of chlorophyll through the "extreme events waves", defining a tool which is able to detect events with a wide range of values of the variable, as guaranteed by the site-specific threshold. The heterogeneity of the maxima of the Mediterranean surface chlorophyll is captured by our severity index, which shows the highest values in northwestern Mediterranean Sea and the lowest ones in southern Levantine Sea. The fact that a single relatively strong event in the oligotrophic region can cover the whole p99 is a result of the analysis, which highlights the heterogeneity of Mediterranean dynamics.
Moreover, we would like to point out that the time series of surface chlorophyll in northwestern Mediterranean Sea (NWM) do not necessarily overshoot the p99 threshold in each year. We report in left panel of Fig. R1.2 an example of local time series of surface chlorophyll in NWM, which shows that the local threshold is not overcome in 5 of the 19 years (i.e., 1995, 1996, 2007, 2009, 2012):

[Figure]

**Fig. R1.2** Time series of surface chlorophyll in a site belonging to northwestern Mediterranean Sea, with daily climatology computed in the site (left) and histogram of frequency of the chlorophyll values recorded in the site (right). In both panels, the horizontal dashed line indicates the 99th percentile threshold computed in the site.

Thus, we suggest that our method identifies extreme events that not necessarily are blooms (i.e., in the oligotrophic areas) and that not all blooms are necessarily extreme events (as shown in Fig. R1.2).

The text and the title are confusing. If you are characterizing blooms (or maxima in chl maximum) using an extreme event method, it is great, but present it like this. Please, don't try to oversell it. A title like "Tracking the Mediterranean blooms using Extreme event waves method" or something like that. Of course the current title is more punchy, but personally, when i read it i've imagined dozens of possible things.

Thank you for this comment. Since we will exclude the reference to "blooms" and we think that the validity of the model is more general than our specific application to the surface chlorophyll (i.e., it can be applied to integrated chlorophyll, to other ecosystem variables, and to both time series as derived by the model output and deseasonalized time series), we propose to modify the title as:
*Extreme events waves in marine ecosystems: an application to Mediterranean Sea surface chlorophyll*

Also, make it clear in the text :
-p1 l14: identify the maxima of chlorophyll as exceptionally high and prolonged "blooms"
We propose to modify it as:
*identify the maxima of surface chlorophyll distribution as continuous and prolonged "waves" of events*

-p2 l54: This allowed to identify maxima of phytoplankton blooms (Desmit et al., 2018), but also positive anomalies with values too low to be actually considered "bloom""
We propose to modify it as:
*This allowed to identify the maxima of surface chlorophyll distribution, which can correspond to phytoplankton blooms (as in Desmit et al., 2018), but also to positive anomalies with values too low to be actually considered "blooms""*

-p6 l172 : (i.e., exceptionally high and prolonged "blooms", as clarified in Introduction)
We propose to modify it as:
*(i.e., continuous and prolonged "waves" of events, as clarified in Introduction)*

-p9 l272 : probably the clearer explanation : "we propose a new method to tackle extreme events in the marine ecosystems on the basin scale. The method is then applied to the surface chlorophyll in Mediterranean open-sea areas to investigate maxima in the winter-spring blooms".
We  propose to modify it as:

*We propose a new method to tackle extreme events in the marine ecosystems on the basin scale. The method is then applied to the surface chlorophyll in Mediterranean open-sea areas to investigate the maxima of chlorophyll distribution in the winter-spring period.*

So sometimes it is "exceptionally high and prolonged", some other time it is "maxima in the winter-spring" blooms. The second (which includes the first) sounds more accurate, but read both is confusing. please make it clearer.

Thank you for the comment. We have just replied point by point in a consistent way.

What struggles me is the lack of definition for bloom and for bloom maxima, Or at least what you consider "blooms" and "bloom maxima" in this study. what gives me the impression of not being sure of what we are looking for, and results with places where a bloom maxima appears every year, and other places where it happens once or twice in 18 years and last 90 days. In the oligotrophic region, where there is no blooms, an EEW is found by construction (as said p6,l181 : "Considering the temporal extension of the simulation (approximately equal to 7000 days), the number of POTs in each grid point is by construction equal to 70.") the long EEW might well be an eddy with higher surface chl concentration inside. It cannot be considered a bloom.

We agree that the use of term "bloom" was misleading in the previous version of the manuscript. As already replied to previous comments, we will avoid the use of term "bloom" in the new version of the manuscript and we will define more precisely the maxima of chlorophyll distribution as our "extreme events" at each point (see also the second next point).
However, we would like to specify that in each point we find by construction 70 Peaks Over Threshold (POTs), since our threshold is equal to 99th percentile and we use a simulation of daily chlorophyll for 19 years. The occurrence of an EEW is instead restricted to the chance of occurrence of POTs in consecutive times and neighbouring points (i.e., spatio-temporal contiguity of extreme events). Additionally, we imposed further criteria on duration (at least 2 days, to avoid possible transient spikes) and area (greater than 4 $\Delta$x × 4 $\Delta$y, with $\Delta$x, $\Delta$y grid spacing in the zonal and meridional direction, respectively, Grasso 2000). Not all POTs obtained are thus included in a EEW (the percentage of POTs not included in EEWs is equal to 0.5%). Conversely, an EEW can include different POTs recorded in the same point at different times.
Moreover, for sure there are EEWs also in oligotrophic regions, since the thresholds are site-specific, and we observed also long EEWs in these regions. We will avoid the term "bloom" in those cases, but the obtained results are valid anyway.

A solution could be to:
– Stop talking about blooms for the whole Mediterranean sea. It would make more sense if you were talking of "(...) investigate maxima in the winter-spring surface chlorophyll maximum". That would be more correct, the maxima being not necessarily extreme, and not saying the word "bloom" don't mislead the attention on something specific that does not occur everywhere in the Mediterranean sea.

We thank the reviewer for this suggestion: we propose to avoid the term "bloom" and to specify the expression "maxima of chlorophyll distribution" in the reviewed version of the manuscript. Nevertheless, these maxima are strictly our "extreme events" at each point, i.e., the values over the 99th percentile threshold at each point. Please see our general reply to the Overall Comments.

– And stop talking about extremes everywhere. The method you use is a method that is first made to find extreme events, but the way you use it, you don't only find extremes. an extreme event that comes back at least once a year is not an extreme, it is part of the normal annual cycle.

As we have already replied to the Overall Comments, we consider maxima of chlorophyll distribution (i.e., our "extreme events") independently from their spread over the years. The inter-annual variability is evaluated in retrospect, on the EEWs, by means of the anomaly index and, as pointed out previously, it highlights the great heterogeneity of the Mediterranean Sea.

We propose to add new lines in the section 2.1.1 of the manuscript, at line 84:

*Extreme events are thus represented by the maxima of the variable distribution in the (x,y) point. They are "rare" events, selected independently from their distribution over the years, in a number equal to 1% of the total records.*

and to substitute line 284 in the Discussion section with:

*In our specific application, it is noteworthy to specify that maxima of surface chlorophyll distribution (i.e., the extreme events, as defined in Sect. 2.1.1) do not necessarily correspond to "blooms" (Siokou-Frangou, 2010), since the extreme events are identified in all points of the domain, including the oligotrophic areas. Moreover, these maxima of chlorophyll are not necessarily distributed in a regular way over the years, due to the inter-annual variability of chlorophyll time series.*

*Our method is able to characterise intensity and regularity of the extreme events in retrospect, by means of mean severity and anomaly indexes computed on the EEWs.*

*In particular, the mean severity index associated to a chlorophyll EEW can be ...*

– Something else that could help to better visualise how extreme the EEW are. You could try to plot the surface Chl annual cycle (with STD in dashed line) for each Mediterranean regions (Fig 3), with the averaged 99th percentile threshold represented on top. that way we can appreciate how "extreme" an EEW is for each area (Maybe you want to adapt the area so it looks more like the fig 6 ? might be more relevant).

Thanks for raising this point.

In the present reply we show examples of local time series of surface chlorophyll in Ionian Sea and northwestern Mediterranean Sea, in Figs. R1.1 and R1.2, respectively. These plots show how "extreme" are the POTs in the selected sites, through the chlorophyll values with respect to the thresholds and the inter-annual variability (i.e., the regular/irregular occurrence of POTs over the years). The plots show also how the daily climatological series computed in the selected sites are well below the p99 thresholds in the sites.

Moreover, a figure reporting the p99 spatially averaged over regions, superimposed to the spatially averaged annual cycle, would mislead the p99 meaning, since p99 is defined locally. A comparison among different areas which is based on spatial means would not have a direct link with the EEWs indexes and would give information only about the heterogeneity of (mean) surface chlorophyll in the considered areas.

Understanding and thus quantifying how "extreme" are the EEWs in the different Mediterranean areas (e.g., their mean severity, anomaly, duration, uniformity) can be instead done from Fig. 5 and Table 2 of the manuscript.

We propose not to add figures like R1.1 and R1.2, since they do not add information needed for the method illustration. Nevertheless, if Reviewer#1 think that figures like R1.1 and R1.2 are important, we would add one figure for each area of Figure 3, in a new Appendix of the revised manuscript.

Unless you want to talk about extremes and only extremes. Then you have to adapt the threshold by taking into account the surface chlorophyll annual cycle as suggested above.

We do not agree with this point, since we think that our study tackle "extreme events" consistently from a statistical point of view, as rare events corresponding to the upper tail of the distribution of

the chosen variable, as the highest values recorded in a certain time period. Moreover, conducting the same analysis on the deseasonalized time series (which is anyway possible within our scheme) would give different results, which answer to a different scientific question. Please see also our reply to the Overall Comments.

Apart from this (important) semantic question, the method is nice and prove to be able to identify, characterize and track the EEW beautifully.
– Also, talking about extremes, i wanted to rise a question, just for discussion. I understand the choice of surface Chl maxima is mainly to test the method and show how it works. But thinking about Mediterranean sea, climate change and extreme events, i wonder if tracking maxima of surface chlorophyll maximum is what i would do. I don't think we can get hypoxia or eutrophication with 12th degree model, this is rather a coastal and river mouth problem. We know that a climate impact could be to lower the deep water formation and hence the bloom. We could use your method (adapting the threshold, considering the Annual cycle) to track years with little or no bloom, and understand why, and see the trends. Or in summertime if your model include dust deposition on high frequency, see if the model shows EEWs linked to dust deposition events,... There is lots of other application of your methods that could make lots of sense (Lots of nice study in perspective).

Thank you for this meaningful observations.
As a first application of the method, we chose the surface chlorophyll since: it is representative of the marine ecosystem functioning; it has been widely investigated in previous studies; model simulation is comparable also with remote sensing measurements (as done in Sect. 3.1, Fig. 4), which increases the confidence level on our model-derived results. These reasons make the chlorophyll a good choice to show how the method works, as you wrote.
However, we agree that there are a lot of possible and interesting applications, thanks for your suggestions.
As written in the last part of our reply to the Overall Comments, we propose to explicitly add in Discussion section the possibility to apply our method starting from seasonally varying threshold, with a mention of the anomalies of chlorophyll in summer as example of investigated process.

== **Text remarks**

– I think there are few places where the English could be corrected, but not being a native English myself, i am not the right person to do that. Maybe you could ask a native English around you to double check your manuscript.
Thank you for the suggestion. We will send the new version of the manuscript to an English Editing Service.

– from p5.l134 and all units following : double check the units the -2 and -1 should be up, if you write with latex, you should write kgˆkmˆ{-2} dayˆ{-1}
Thank you. We will correct these typos.

– p6 l73 : " chlorophyll as a proxy for the phytoplankton biomass" Surface chlorophyll is representative of the surface biomass (probably why one of your idea in the discussion is to check the event in 3D)
We will correct the expression by:
*surface chlorophyll as a proxy for surface phytoplankton biomass*

– p6 l82 to 85: "Mapping the 99th percentile threshold values computed at each grid point on the whole basin (Fig. 3), it can be noticed that grid points that are near in space exhibit small differences in their threshold values and also that different patterns are recognisable in the basin. Hereafter, we

use the abbreviations indicated in Fig. 3 to refer to different Mediterranean regions" – So the 99th percentile is fixed in time. This means that you compare toward an ~annual 99th percentile threshold of Chl. basically you will only have EEW during the bloom period. A summer with exceptional summertime Chl will not appear with this method as it will never exceed bloom period values. can't you do a time varying 99th percentile threshold to be able to see non- bloom period EEW ? otherwise you will probably miss the most interesting events... probably needs a longer run to get enough data to keep it statistically feasible.

We are aware that a percentile threshold which is fixed in time allows to recognise only the highest values of chlorophyll in the considered time period, and in this paper we focus on those events.
We do not exclude to investigate also fall blooms, or high anomalies of chlorophyll in summer, but these are different processes, and would deserve separate papers. Please see also our reply to the Overall Comments.

– p7 l191 : " The model-derived chlorophyll patterns (Fig. 4, second column) are in good agreement with the remote sensing data (first column) in the same temporal interval of the EEW" – Hard to tell, seems the sat Chl has a more extended bloom than the model, and starts slightly later (and probably ends later as well). But both model and sat presents an EEW on the same period, what is already a nice model performance ! And you have a nice bloom in the Ligurian sub-basin, that's impressive! Talking about e Ligurian bloom, it does not appear in the EEW area. it is considered as a separated EEW ?

Thank you for this comment. We think to substitute lines 191-193 "The model-derived chlorophyll patterns (Fig. 4, second column) are in good agreement with the remote sensing data (first column) in the same temporal interval of the EEW. In fact, a strong increase…" with:
*Both model and remote sensing data show patterns of high values of chlorophyll in the period of EEW occurrence (second and first column of Fig.4, respectively). A strong increase…*
However, the high values recorded in the Ligurian Sea are included in another EEW. We think to add this information at (old) line 204 as:
*The high values of chlorophyll recorded in the Ligurian Sea on 20th March are associated to a separate EEW (not shown).*

– p8 l235 : " are around half of the ones of ALB or NWM." needs to be rephrase.
We think to rephrase it as:
*are about half of the values displayed by ALB or NWM*

– p8 l239 : "with a similar chlorophyll EEWs phenomenology."
We think to rephrase it as:
*with a similar phenomenology of chlorophyll EEWs.*

– p10 l288 : "pointed out the heterogeneity of the blooms intensity in the Mediterranean Sea" - back to my main comment, you don't see blooms everywhere...
As we replied to the Overall Comments, we will avoid the use of the term "bloom" in the new version of the manuscript. We think to substitute that expression with:
*pointed out the heterogeneity of the maxima of chlorophyll in the Mediterranean Sea*

– p10 l310 : Furthermore ?
Thank you for this correction. We will substitute "further" with
*furthermore*

– p10 l310 : you could have shown the "spatio-temporal persistence", it looks like a nice index. Why not show it ?

We did not include the map of the spatio-temporal persistence in the previous version of the manuscript because this additional index is directly obtained as the product of two indexes, more meaningful in our opinion, which have been already shown in Fig. 5: uniformity, i.e., (spatial) persistency of the EEW, and duration, i.e., total time of occurrence of the EEW.
We recognise that this index should have been defined in Sect. 2.1.2 before its mention in Discussion, but we prefer to avoid it, to not weigh down the manuscript. Therefore, we would like to delete the sentence at lines 310-311.

– p12 l360 to 374 : Good idea!

Thank you for the comment.

– p12 l375 : "A critical parameter of our method is the choice of the local percentile threshold" - I agree looks like one of the key of the method. but why this choice of a simple percentile threshold, and not include the local annual cycle ( maybe + a*STD ) in the threshold (As i mentioned above depending what you want to analyse, it can be justified, it can be the right choice) ?

We agree that the choice of the threshold depends on the scientific question. Please see our reply to the Overall Comments.

– p13 l 197 : "Of the clusters with the highest content of all the indexes has been generally maintained both in case of higher and lower thresholds" - rephrasing : of the clusters with the highest index values,...,
Thank you. We will follow this suggestion in the reviewed version of the manuscript.

– p13 l396 : "A key issue" - not issue, it is one of the strength of this method, not issue i think, and from all what you could do in your study because of that.
We agree with your comment. We will rephrase "key issue", by using:
*key point*

– p13 l400 : "The time series in the grid point" - rephrase : Each grid point's time serie
We will rephrase it.

– p13 l400 : " allowed to maintain a definition of "extreme" relative to the local ecosystem properties."
This i do not agree. in some places like the most oligotrophic regions, you probably find extremes, but in the bloom regions, it is not.
We do not agree with this observation. Also the time series recorded in points belonging to bloom regions (e.g. Gulf of Lion) can show an inter-annual variability such that the 99th percentile threshold is overcome only in some years, as shown in Fig. R1.2 in the first part of this reply.

–P23-Fig4–You talk about the MLD in the text,but you don't show it on the plot. Of course, we can guess the Mixed layer is very deep where the NO3 is high and Chl low, but, it might be good to add iso-contour with depth values on the Chl plot for example. That would help both the writer and the reader. – very nice bloom in the Legurian sub- basin! You must have a very high res atm model with high freq coupling. you should add these details in the model description. It help understand the results.
In the reviewed version of the manuscript we will add further details related to the atmospheric forcing in the model description, at line 155:

*The atmospheric fields used to force the simulation come from a 12 km horizontal resolution regional downscaling of ERA-Interim reanalysis (Llasses et al., 2016; Reale et al., 2017) and drive the simulation every 3 hours.*,

thanks for the suggestion.

With regard to MLD, we have already plotted its time series in three points, internal, peripheral and external to the EEW in Appendix (Figs. A.2-A.4) to help the interpretation of the results.

We recognise that including MLD also in Fig. 4 can help the reader, but we are worried about the readability of the new resulting figure, since Fig. 4 is already full of detailed information. Thus, we propose to keep Fig.4 as it is now and to refer to Figs. A.2-A.4 for MLD evolution.

– P24 - Fig 5a – Difficult to interpret.... and the color-scale does not help. how many EEW occur per year ? how many on the hole period ? I don't understand what you mean here.

We recognise that Fig. 5a in the manuscript needs graphic improvement. It represents in each grid point the probability of occurrence of more than one EEW per year.

However, considering Reviewer#1's comment, we decided to modify it, by representing more simply the total number of EEWs in the period 1994-2012. Therefore, we propose to substitute Fig. 5a with the following (Fig. R1.3):

[Figure]

**Fig. R1.3** - Proposed reviewed Fig. 5a.

and the first part of the Fig.5 caption accordingly:

*Fig.5: Number of surface chlorophyll EEWs occurred in 1994-2012 (a) and means of the indexes of chlorophyll EEW occurred in the Mediterranean Sea in 1994-2012:...*

Moreover, we propose to modify the text referred to Fig. 5a, in Sect. 3.2.1 at lines 224-227, as:

*Figure 5 displays the total number of EEWs occurred in each Mediterranean point (Fig.5 a) and the mean values of the EEW indexes, computed as the mean of indexes of all the EEWs that involved that point (Figs. 5b-f). Since there are Mediterranean areas showing more than one EEW per year (as can be inferred from Fig. 5a), the initiation time in each grid point and year was associated to the most severe EEW of that year.*

and in Discussion, at lines 313-314:

*The initiation index was excluded from the computation since there are areas of the basin showing more than one EEW per year per grid point (Fig. 5a).*

– P27 - Fig A.2, A.3, A.4 – I cannot do the difference between the climatological line end the 2005 one. Please, try with different dashed or doted line to find one that we really can see.
Thank you for your comment. We will improve the graphic quality of the Figs. A.2, A.3, A.4 in the new version of the manuscript, as reported here for the new Fig. A.2 (Fig. R1.4). The other figures will be modified accordingly.

[Figure]

**Fig. R1.4** - Proposed reviewed Fig. A.2.

Bibliography

Llasses, J., Jordà, G., Gomis, D. et al. Heat and salt redistribution within the Mediterranean Sea in the Med-CORDEX model ensemble. Clim Dyn 51, 1119–1143 (2018). https://doi.org/10.1007/s00382-016-3242-0.

Reale, M., Salon, S., Crise, A., Farneti, R., Mosetti, R., & Sannino, G. (2017). Unexpected covariant behavior of the Aegean and Ionian Seas in the period 1987–2008 by means of a nondimensional sea surface height index. Journal of Geophysical Research: Oceans, 122, 8020– 8033, https://doi.org/10.1002/2017JC012983.

---

## Author Comment (AC3) · 26 May 2020

We sincerely thank Reviewer#2 for his/her comments, which gave us the opportunity to deepen some points of our paper. We indicate here our reply in blue colour and some corrections we propose to implement on the text of the manuscript in italic red. The full version of our reply, if accepted, will include all the corrections declared here.

The manuscript "Characterisation of extreme events waves in marine ecosystems: the case of Mediterranean Sea" describes a new method to characterize extreme events bases upon simulated chlorophyll concentrations for the period 1994-2012. Using a cluster analysis applied to a set of indices that define the occurrence of extreme events waves, different ecosystem regimes were defined. In my opinion, the manuscript is very interesting and deserves publication in Biogeosciences after minor to moderate revisions. In detail:

1) The language should be checked by a native English speaking person. There are several typos that should be removed. For instance in line 270 there is a reference to Fig. 9 that does not exist.

Thank you for the suggestion. We will carefully revise the text references to Figures and Tables and we will send the new version of the manuscript to an English Editing Service.

2) The method uses surfaces chlorophyll concentrations averaged over the uppermost 10 m and does not consider vertical profiles. Please discuss why subsurface blooms do not play a role.

We considered surface chlorophyll particularly suitable to show the functioning of the method, since surface chlorophyll has been widely investigated in literature and it is comparable also with remote sensing measurements (as done in Sect. 3.1, Fig. 4). However, also subsurface processes (e.g., associated to the Deep Chlorophyll Maximum feature) can be analysed by our method, using the same model implementation (which provides 3D chlorophyll concentration fields). For sure, it can be a further interesting application of our method and we will specify it in the new version of the manuscript, in the Discussion section.
In particular, we propose to substitute the two sentences at lines 360-364 with:
*We have applied the method to the surface chlorophyll, as one of the more representative and investigated variables of the marine ecosystem, identifying maxima of chlorophyll as events which potentially influence the ecosystem function (e.g., food web and carbon fluxes). However, our method can be applied also to integrated chlorophyll, to account for subsurface growth of phytoplankton, or to other variables whose impacts on the ecosystem can be relevant, such as HAB-like phytoplankton groups (Vila and Masó, 2005), provided the availability of a continuous dataset in time and space.*
*The method may also be applied to other specific subsurface quantities, as for example the surface identifying key biogeochemical properties as the Deep Chlorophyll Maximum (e.g. as defined in Salon et al., 2019), the oxygen minimum or the oxygen deficiency (OSPAR, 2013; Ciavatta et al., 2016).*
Anyway, as replied also to Reviewer#1, we decided to avoid the reference to "bloom" processes in the new version of the manuscript, and to restrict our argumentation to extreme events (i.e., maxima of surface chlorophyll distribution).

3) It is discussed but I am still worried about the stability of the identified regimes using different thresholds. You showed that the clustering of the mean values of the indices do not change much. However, are there changes in the spatial distribution of the regimes shown by Figure 6?

Thank you for this comment, which gives us the opportunity to deepen some aspects.

We applied the fuzzy k-means analysis also in case of 98th and 99.5th percentile thresholds and we report the results in Figs. R2.1 and R2.2, respectively.

[Figure]

**Figure R2.1** Fuzzy clusters with maximum membership, in case of 98th percentile threshold.

[Figure]

**Figure R2.2** Fuzzy clusters with maximum membership, in case of 99.5th percentile threshold.

From a general point of view, the clusters distribution for 99.5th percentile threshold (Fig. R2.2) is very similar to the clusters distribution for 99th percentile (Fig. 6 of the manuscript), while the clusters distribution for 98th percentile (Fig. R2.1) differs mainly in Ionian and Levantine Sea.

More in details, the spatial distribution of clusters #3, #6, #7 in the western basin does not considerably change with respect to Fig. 6 of the manuscript, as well as the spatial distribution of cluster #4 in the eastern basin.
Clusters #1, #2, #3 #5 and #6 display instead differences, mainly in the clusters distribution for 98th percentile, in the eastern basin, i.e., southern Adriatic Sea, central Ionian Sea, Aegean Sea, Libyan coast and eastern Levantine Sea.

Since the identification of clusters depends on four indexes (i.e., mean severity, uniformity, duration, anomaly) which do not necessarily scale in the same way with the local threshold, the fact that the spatial distribution resulting from the fuzzy k-means analysis (as "combination" of the four indexes) for different thresholds can differ from Fig. 6 is a reasonable result.
Moreover, it should be highlighted that Figs. 6, R2.1, R2.2 show the clusters referred to the maximum membership.
In addition to maximum membership, we considered also the "confusion index" (Burrough et al., 1997), which, applied to our case, quantifies how well each point of the Mediterranean domain has been classified. High values of the confusion index (CI) index are related to higher sensitivity of some areas in the cluster classification with respect to variations in the local threshold (i.e., to differences of Figs. R2.1, R2.2 with respect to Fig. 6).
We estimated the confusion index (CI) as:
$CI=1-(MF_{max}-MF_{max2})$,
where $MF_{max}$ denotes the dominant membership value and $MF_{max2}$ is the subdominant membership value for each point, and we computed it in case of 99th percentile threshold. Figure R2.3 shows values of CI greater than 0.7 (i.e., "high values" of CI) as black dotted points.
We can observe that most of the areas displaying differences in the spatial distribution of the clusters with respect to variations in the threshold (Figs. R2.1, R2.2) correspond to high CI in the reference case (Fig. R2.3) and that the identification of the clusters generally appears to be consistent with the other two clusterizations.

We recognise that this point deserves a revision in the manuscript.
Therefore, we propose to substitute Fig. 6 with Fig. R2.3 in the revised manuscript, adding the expression:
*with black points indicating a confusion index higher than 0.7*
at the end of the figure caption and the definition of the confusion index in the text. Accordingly, our statement about the robustness of the classification (old lines 387-390) will be reformulated and referred to this new version of Fig. 6.

[Figure]

**Figure R2.3** Fuzzy clusters with maximum membership, in case of 99th percentile threshold, with black points indicating a confusion index higher than 0.7.

4) In Section 4 and the Appendix, ecosystem dynamics characterizing some of the regimes are discussed. However, some clusters lack any dynamical explanation and might be rather artificial. It would increase the scientific value of the manuscript if you could discuss these clusters more in detail as you have done it for NWM in the Appendix.

Section 3 and Appendix of the first version of the manuscript present in detail one of the Extreme Event Waves (EEWs) identified in the 1994-2012 dataset, to show how the method works. In particular, Appendix highlighted that the method catches all and only the relevant information of the event. We selected the EEW associated to the highest value of mean severity of the whole dataset (and occurred within the area covered by cluster #7) as a particularly meaningful case.
Since the main focus of the paper is to propose a method to identify and classify EEWs, rather than to analyse in detail surface chlorophyll dynamics, we did not include other examples of EEWs (e.g., associated to areas covered by other clusters, as suggested) in the previous version of the manuscript, avoiding to enlarge too much the length of the manuscript and to shift the attention more on the specific application than on the method. Thus, we would prefer not to add new figures and comments in Appendix.
However, if the Reviewer#2 still suggests to do it, we are open to include other examples of EEWs occurring in areas covered by other clusters.

5) In the abstract you mentioned that "There is a growing interest about events that can affect ecosystem functions and services in a changing climate". However, is the method suitable for following the temporal shifts in the regimes without any discontinuities? I suggest that you split the period in half and that you use the cluster analysis for both periods. By this, the impact of trends in some of the indices on the spatial distribution of the regimes might be investigated and compared with observed changes.

Thank you for the comment.
Yes, our method is suitable for following the temporal changes in the characteristics of the EEWs, provided that the time series is long enough for a robust trend or regime shift evaluation on a multidecadal scale. Unfortunately, our time series is not very long (i.e., 19 years). Thus, we think that the approach to split the chlorophyll dataset in two parts, i.e. 9-10 years for each part, would not guarantee statistical robustness to the analysis.

Indeed, we conducted a non parametric analysis of the trend slope (Theil-Sen method) and showed the results in Tab.2 of the manuscript, by reporting the cases in which the ratio between slope and mean is higher than 1% (red cells) and lower than -1% (blue cells). For sake of clarity, we report here the annual time series (and trend slope) of each cluster for the duration, uniformity, mean severity and anomaly indexes (Figs. R2.4-R2.7). As it can be intuitively grasped, the length of the time series does not allow further speculations on temporal changes of the EEWs characterisation and classification.

[Figure]

**Figure R2.4** Trend evaluation (as slope over mean) on the clusters in Fig.6, referred to duration index.

[Figure]

**Figure R2.5** As Fig. R2.4, but referred to uniformity index.

[Figure]

**Figure R2.6** As Fig. R2.4, but referred to mean severity index.

[Figure]

**Figure R2.7** As Fig. R2.4, but referred to anomaly index.

Bibliography:

Burrough, P. A., van Gaans, P. F. M. and Hootsmans, R.: Continuous classification in soil survey: spatial correlation, confusion and boundaries, Geoderma, 77(2), pp. 115–135. doi: https://doi.org/10.1016/S0016-7061(97)00018-9, 1997.

Ciavatta, S., Kay, S., Saux-Picart, S., Butenschön, M., and Allen J. I.: Decadal reanalysis of biogeochemical indicators and fluxes in the North West European shelf-sea ecosystem, J. Geophys. Res.-Oceans, 121, 1824–1845, https://doi.org/10.1002/2015JC011496, 2016.

OSPAR: Common procedure for the identification of the eutrophication status of the OSPAR maritime area, Tech. Rep. 2013-8, London, UK, available at: https://www.ospar.org/documents?d532957 (last access: 28 October 2019), 2013.

Salon, S., Cossarini, G., Bolzon, G., Feudale, L., Lazzari, P., Teruzzi, A., ... and Crise, A.: Novel metrics based on Biogeochemical Argo data to improve the model uncertainty evaluation of the

CMEMS Mediterranean marine ecosystem forecasts. Ocean Sci., 15(4), 997-1022 https://doi.org/10.5194/os-15-997-2019, 2019.

---

## Author Response (AR1)

In this document we include our replies to Reviewers' comments and the new version of the manuscript, with the corrections reported in red.

The English language has been revised by a professional service and we include the certificate at the end of the document.

Please note that the minor modifications to the English language are not highlighted in red, in order to facilitate the reading of the new manuscript.

In particular, in the new version of the paper we provided a clearer definition of "extreme event" and "extreme event wave (EEW)" (Sect. 2.1.1) and we carefully revised the use of the terms "blooms" and "maxima" throughout the text (please see our reply to Reviewer#1's Overall Comments), with also a further clarification with respect to our previous reply to Reviewer#1's comments posted in the Biogeoscience open discussion.

Moreover, we extended the proposed method to any ecosystem variable (Sect. 2.1) and we revised the Discussion section related to the possible applications of the method (also by reorganizing the order of the arguments). In particular, we explicitly added in the text the possible application to seasonally varying thresholds (as thoroughly discussed in our reply to Reviewer#1's Overall Comments) and we accounted for Reviewer#2's comment #2 about chlorophyll in subsurface layers.

Furthermore, we revised the part of the manuscript concerning the stability of the identified regimes of surface chlorophyll EEWs and we proposed a new version of Fig. 6, that accounts for the "confusion index" computed in the fuzzy clustering analysis (see our reply to Review#2's comment #3).

Finally, we also provided new versions of Fig. 5a (please see our reply to Review#1's comment: P24 – Fig 5a) and Figs. A.2-A.4 (following Review#1's suggestion: P27 - Fig A.2, A.3, A.4) and we modified the format of Figs. 3-4-5-7 and Fig. A.1.

\*\*\*\*\*\*\*\*\*\*\*\*\*\*\*\*\*\*\*\*\*\*\*\*\*\*

**Reply to Reviewer#1's comments**

We sincerely thank Reviewer#1 for his/her comments, which gave us the opportunity to clarify some key points of our paper.

We indicate our reply in blue colour and the corrections of the manuscript text in italic red.

The English language has been revised by an Editing Service, as proved by the certificate at the end of this document. Some comments referred to single English terms were not strictly accepted only because the provided translation reformulated the corresponding sentences.

As regards our previous reply to Reviewer#1's comments posted in the Biogeoscience open discussion, we adopted in this later version an additional revision of the expressions "maxima of local distribution" and "maxima of chlorophyll" to further clarify the terminology. In particular, we dropped the term "maxima" in those cases and we strictly referred to "extreme events" as the values over the 99th percentile threshold in the point, i.e., top 1% values of the local distribution.

**== Overall Comments**

In this Study, Valeria Di Baggio et al. use an extreme event identification method to track the late winter-early spring blooms in the Mediterranean sea. Their method enable to identify and follow day by day the bloom propagation, and characterize the event with different indexes.

Although the method is shown to be powerful and useful, I have some questions/concern with the application done here with the Mediterranean surface chlorophyll, as I am not sure what we are looking for, and getting in the end... Are we looking for extremes ? blooms ? strong blooms ? blooms maxima ? maxima of surface chl maximum ? we are not sure, and the way it is done probably allow all of those. But then... Are blooms considered as extreme events ?

This study aims to propose a methodology for the analysis of "extreme event waves" (EEWs), defined as a set of "extreme events" which are contiguous in space and time.

We defined statistically an "extreme event" in each point (x,y) as a value of the variable which is over a given threshold. In our case, the threshold is set as the 99th percentile of the time series recorded at that specific point (i.e., the threshold is site specific: p99 = p99(x,y)).

Extreme events are thus represented by the top 1% values of the variable distribution in that point (i.e., upper tail of the local distribution). They are "rare" events (i.e., their number is equal to 1% of the total records; right panels in Fig. R.1.1 and R1.2), selected independently from their distribution over the years. In fact, the temporal regularity over the years (i.e., the inter-annual variability) is one of the features which, in retrospect, can be quantified in the analysis of the "extreme event waves" by means of our "anomaly" index.

In our specific application focused on surface chlorophyll, the "extreme events" are strictly identified by the values of the local distribution of surface chlorophyll that are above the 99th percentile threshold, i.e., the top 1% values of the surface chlorophyll distribution. The occurrence of these values can be distributed quite regularly over the years and thus correspond to annual maxima (i.e., in the seasonal cycle), or can spread among a few years, as in case of the northern Ionian Sea (as shown in Fig. R1.1, below). In the latter case, such high inter-annual variability has been detected in the EEWs covering that area, by means of high values of the anomaly index.

On the other hand, the extreme events of surface chlorophyll can correspond to blooms (e.g., in the Gulf of Lion), but also to chlorophyll values which are too low to be properly considered as "blooms" (e.g., in the southern Levantine Sea). Our method is able to distinguish between these two cases, by means of high and low values of the severity index of the EEWs, respectively.

[Figure]

**Fig. R1.1** Time series of surface chlorophyll in a site belonging to northern Ionian Sea, with daily climatology computed in the site (left) and histogram of frequency of the chlorophyll values recorded in the site (right). In both panels, the horizontal dashed line indicates the 99th percentile threshold computed in the site.

Thus, considering Reviewer#1's objection, we propose a method to analyse the extreme events; based on our analysis, blooms can be extreme events, but not all extreme events can be classified (generally speaking) as blooms.

We recognise that our use of the terms "bloom" and "maxima" (of chlorophyll/of blooms) in the previous version of the manuscript was quite confusing, and we have avoided these terms in the revised manuscript.

Apart from this main and i think important concern, the study is nice and relevant. the way the authors manage to track and characterize these events is shown to be useful, with lots of relevant information, and could be exported for all kind of extreme event study.

I really appreciate this study, but it has to make clear what we are looking at: extreme? or surface chlorophyll maximum ? depending on the answer, the amount of work needed to correct the paper will be different, corresponding to a major review if you want to make it an extreme event analysis; or a minor review if it rather is a surface chl maximum analysis using an extreme event tool (what i think the authors are doing here).

As previous clarified, we actually conducted an analysis on the top 1% values of the surface chlorophyll distribution (i.e., "extreme events", with respect to p99(x,y) of the local chlorophyll distribution) and the subsequent steps of the method (i.e., the identification of the extreme event waves (EEWs), their characterisation and classification) can be defined an "extreme events tool", as Reviewer#1 suggests.

Nevertheless, we hold that the validity of the whole method is general, i.e., the same definitions can be applied also to deseasonalized time series (see also the second next point). In that case, the "extreme events" under study would be identified by the values differing the most from the daily climatological mean (where "the most" would be set by the 99th percentile threshold of the distribution of the anomalies). The "extreme events" identified in this latter way do not necessarily correspond to the highest values of the variable recorded in the point and identify local perturbations with respect to the seasonal cycle. The choice to deseasonalize or not the time series depends on the scientific question.

In our analysis, we focused on top 1% values of surface chlorophyll distribution as the highest values recorded in the time series and limited their characterisation and classification to the winter-spring period.

We recognise that the difference between absolute and time-dependent thresholds, as well as the possibility of following the latter approach in the first part of our method, deserves an explicit mention in the manuscript. Thus, we added the following sentences in the Discussion section, at lines 415-419 (old line 393):

*Finally, our method of EEW identification, characterisation and classification can also be applied to extreme events that are defined starting from seasonally varying threshold (as e.g. in Hobday et al. 2016). In this case, "extreme events" would correspond to the highest anomalies recorded with respect to the climatological seasonal cycle of the variable and generally not to the highest values of the variable recorded throughout the time series (as in our case of temporally fixed threshold). Such an application would allow us to investigate different kinds of scientific questions, such as chlorophyll anomalies in summer.*

**== Extreme, Bloom, or Surface Chlorophyll Maximum ?**

Although the method is shown to be powerful and useful, I have some questions/concern with the application done with the Mediterranean surface chlorophyll, as I am not sure what we are looking for extremes ? blooms ? strong blooms ? blooms maxima ? maxima of surface chl maximum ? it seems you see all of those including extremes, like the one you have selected for the example. But then... Are blooms considered as extreme events ?

Please refer to our reply to Overall Comments about the concepts of "extreme", "blooms" and "maxima" (of chlorophyll/of blooms).

From the definition you give (Page 2, line 35) "a large deviation from a reference state", but i think the reference state should include the annual cycle... if you are looking for extreme. If your targets are extremes of surface Chl, you still could use the 99th percentile threshold, and only keep those going above the local annual cycle + STD (or 1.5 * STD) for example, or instead of a 2D threshold make it 3D, including an annual cycle, that could also show extreme in summertime (maybe due to dust events for example),... The choice of a 2D 99th percentile on the whole period is somehow too broad if you look for extremes, and of course you will get completely different results in the different area of the Mediterranean sea. In the North-Western part with strong and spiky blooms,you will overshoot the threshold at least once a year, because of this spiky bloom configuration (but that's not extreme... it is the every year bloom), whereas in the oligotrophic region, where the chl phenology is smooth, a relatively stronger event can cover the whole 99th percentile. So, if you were looking for extreme, you should change that, adapt the threshold which seems to be the key of the method. But i am not even sure you are looking for extreme.

As we replied also to the Overall Comments, we think that deseasonalizing or not the time series is a choice that depends on the scientific question. In our case, we investigated the highest values of surface chlorophyll defined by the tail of the distribution of chlorophyll in each point. The investigation of "extreme events" of anomalies with respect to the daily climatological mean is a different (and, for sure, interesting) study. It would give us results of a different kind, which would deserve a separate paper. However, our method can be applied also to the deseasonalized time series, since it analyses the spatio-temporal contiguity of "extreme events" and provides a "tool" for their characterisation and classification.

In our analysis, we propose a description of the phenomenology of extreme events of chlorophyll through the "extreme event waves", defining a tool which is able to detect events with a wide range of values of the variable, as guaranteed by the site-specific threshold. The heterogeneity of the top 1% values of the local distribution of surface chlorophyll in the Mediterranean Sea is captured by our severity index, which shows the highest values in northwestern Mediterranean Sea and the lowest ones in southern Levantine Sea. The fact that a single relatively strong event in the oligotrophic region can cover the whole p99 is a result of the analysis, which highlights the heterogeneity of Mediterranean dynamics.

Moreover, we would like to point out that the time series of surface chlorophyll in northwestern Mediterranean Sea (NWM) do not necessarily overshoot the p99 threshold in each year. We report in left panel of Fig. R1.2 an example of local time series of surface chlorophyll in NWM, which shows that the local threshold is not overcome in 5 of the 19 years (i.e., 1995, 1996, 2007, 2009, 2012):

[Figure]

**Fig. R1.2** Time series of surface chlorophyll in a site belonging to northwestern Mediterranean Sea, with daily climatology computed in the site (left) and histogram of frequency of the chlorophyll values recorded in the site (right). In both panels, the horizontal dashed line indicates the 99th percentile threshold computed in the site.

Thus, we suggest that our method identifies extreme events that not necessarily are blooms (i.e., in the oligotrophic areas) and that not all blooms are necessarily extreme events (as shown in Fig. R1.2).

The text and the title are confusing. If you are characterizing blooms (or maxima in chl maximum) using an extreme event method, it is great, but present it like this. Please, don't try to oversell it. A title like "Tracking the Mediterranean blooms using Extreme event waves method" or something like that. Of course the current title is more punchy, but personally, when i read it i've imagined dozens of possible things.

Thank you for this comment. Since we have excluded the reference to "blooms" in the revised manuscript and we think that the validity of the model is more general than our specific application to the surface chlorophyll (i.e., it can be applied to integrated chlorophyll, to other ecosystem variables, and to both time series as derived by the model output and deseasonalized time series), we modified the title as:

*Extreme event waves in marine ecosystems: an application to Mediterranean Sea surface chlorophyll*

Also, make it clear in the text :
-p1 l14: identify the maxima of chlorophyll as exceptionally high and prolonged "blooms"
We modified it as:
*identify and characterise surface chlorophyll EEWs*

-p2 l54: This allowed to identify maxima of phytoplankton blooms (Desmit et al., 2018), but also positive anomalies with values too low to be actually considered "bloom""
We modified it as:
*This application allowed us to identify the extreme events of surface chlorophyll, which can correspond to both phytoplankton blooms (Desmit et al., 2018) and positive anomalies with values too low to be actually considered "blooms" (e.g., in the Levantine Sea)"*

-p6 l172 : (i.e., exceptionally high and prolonged "blooms", as clarified in Introduction)
We modified it at line 177 as:
*(i.e., continuous and prolonged "waves" of extreme events)*

-p9 l272 : probably the clearer explanation : "we propose a new method to tackle extreme events in the marine ecosystems on the basin scale. The method is then applied to the surface chlorophyll in Mediterranean open-sea areas to investigate maxima in the winter-spring blooms".
We modified it (at lines 286-288) as:
*We propose a new method to identify and characterise extreme event waves in marine ecosystems. The method is then exemplified by a first application to surface chlorophyll in Mediterranean open sea areas, with specific reference to the winter-spring period.*

So sometimes it is "exceptionally high and prolonged", some other time it is "maxima in the winter-spring" blooms. The second (which includes the first) sounds more accurate, but read both is confusing. please make it clearer.
Thank you for the comment. We have just replied point by point in a consistent way.

What struggles me is the lack of definition for bloom and for bloom maxima, Or at least what you consider "blooms" and "bloom maxima" in this study. what gives me the impression of not being sure of what we are looking for, and results with places where a bloom maxima appears every year, and other places where it happens once or twice in 18 years and last 90 days. In the oligotrophic region, where there is no blooms, an EEW is found by construction (as said p6,l181 : "Considering the temporal extension of the simulation (approximately equal to 7000 days), the number of POTs in each grid point is by construction equal to 70.") the long EEW might well be an eddy with higher surface chl concentration inside. It cannot be considered a bloom.

We agree that the use of terms "bloom" and "maxima" (of chlorophyll/of blooms) was misleading in the previous version of the manuscript. As already replied to previous comments, we avoided the use of these terms in the new version of the manuscript.
On the other hand, we maintained the use of term "POTs" (i.e., peaks over threshold) to indicate the "extreme events" in the section dedicated to the indexes of the EEWs (Sect. 2.1.2) and we also added the expression "the top 1% values" of the local distribution with specific reference to the 99th percentile threshold in the Sect. 2.1.1 (see also the second next point).

However, we would like to specify that in each point we find by construction 70 Peaks Over Threshold (POTs), since our threshold is equal to 99th percentile and we use a simulation of daily chlorophyll for 19 years. The occurrence of an EEW is instead restricted to the chance of occurrence of POTs in consecutive times and neighbouring points (i.e., spatio-temporal contiguity of extreme events). Additionally, we imposed further criteria on duration (at least 2 days, to avoid possible transient spikes) and area (greater than 4 Δx × 4 Δy, with Δx, Δy grid spacing in the zonal and meridional direction, respectively, Grasso 2000). Not all POTs obtained are thus included in a EEW (the percentage of POTs not included in EEWs is equal to 0.5%). Conversely, an EEW can include different POTs recorded in the same point at different times.

Moreover, for sure there are EEWs also in oligotrophic regions, since the thresholds are site-specific, and we observed also long EEWs in these regions. We have avoided the term "bloom" in those cases, but the obtained results are valid anyway.

A solution could be to:
– Stop talking about blooms for the whole Mediterranean sea. It would make more sense if you were talking of "(...) investigate maxima in the winter-spring surface chlorophyll maximum". That would be more correct, the maxima being not necessarily extreme, and not saying the word "bloom" don't mislead the attention on something specific that does not occur everywhere in the Mediterranean sea.

We thank Reviewer#1 for this suggestion. We have avoided the terms "bloom" and "maxima" (of chlorophyll/of blooms) in the reviewed version of the manuscript. Nevertheless, the top 1% values of the variable distribution (i.e., the values over the 99th percentile threshold at each point) strictly identify our "extreme events", as the highest values at each point. Please see our general reply to the Overall Comments.

– And stop talking about extremes everywhere. The method you use is a method that is first made to find extreme events, but the way you use it, you don't only find extremes. an extreme event that comes back at least once a year is not an extreme, it is part of the normal annual cycle.

As we have already replied to the Overall Comments, we consider the occurrence of values above the 99th percentile threshold computed on the local distribution (i.e., our "extreme events") independently from their spread over the years. The inter-annual variability is evaluated in retrospect, on the EEWs, by means of the anomaly index and, as pointed out previously, it highlights the great heterogeneity of the Mediterranean Sea.
We added new lines in the section 2.1.1 of the manuscript, at lines 82-84 (old line 84):
*Extreme events are thus represented by the highest values (i.e., top 1%) of the variable distribution observed in the (x,y) point. These events are "rare" events and are selected from the total records independently from their distribution over the years.*
and we replaced old line 284 in the Discussion section with (new lines 298-304):
*In our specific application, it is noteworthy to specify that the top 1% values of surface chlorophyll (i.e., extreme events, as defined in Sect. 2.1.1) do not necessarily correspond to "blooms" (Siokou-Frangou, 2010) since extreme events are identified in all points of the domain, including oligotrophic*

*areas. Moreover, the top 1% values of chlorophyll are not necessarily distributed in a regular way over the years due to the inter-annual variability of the chlorophyll time series.*
*Our method is able to characterise the intensity and regularity of extreme events in retrospect by means of the mean severity and anomaly indexes computed on the EEWs.*
*In particular, the mean severity index associated with a chlorophyll EEW can be ...*

– Something else that could help to better visualise how extreme the EEW are. You could try to plot the surface Chl annual cycle (with STD in dashed line) for each Mediterranean regions (Fig 3), with the averaged 99th percentile threshold represented on top. that way we can appreciate how "extreme" an EEW is for each area (Maybe you want to adapt the area so it looks more like the fig 6 ? might be more relevant).

Thanks for raising this point.
In the present reply we show examples of local time series of surface chlorophyll in Ionian Sea and northwestern Mediterranean Sea, in Figs. R1.1 and R1.2, respectively. These plots show how "extreme" are the POTs in the selected sites, through the chlorophyll values with respect to the thresholds and the inter-annual variability (i.e., the regular/irregular occurrence of POTs over the years). The plots show also how the daily climatological series computed in the selected sites are well below the p99 thresholds in the sites.

Moreover, a figure reporting the p99 spatially averaged over regions, superimposed to the spatially averaged annual cycle, would mislead the p99 meaning, since p99 is defined locally. A comparison among different areas which is based on spatial means would not have a direct link with the EEWs indexes and would give information only about the heterogeneity of (mean) surface chlorophyll in the considered areas.
Understanding and thus quantifying how "extreme" are the EEWs in the different Mediterranean areas (e.g., their mean severity, anomaly, duration, uniformity) can be instead done from Fig. 5 and Table 2 of the manuscript.

We decided not to add figures like R1.1 and R1.2, since they do not add information needed for the method illustration. Nevertheless, if Reviewer#1 think that figures like R1.1 and R1.2 are important, we propose to add one figure for each area of Figure 3, in a new Appendix of the revised manuscript.

Unless you want to talk about extremes and only extremes. Then you have to adapt the threshold by taking into account the surface chlorophyll annual cycle as suggested above.

We do not agree with this point, since we think that our study tackle "extreme events" consistently from a statistical point of view, as rare events corresponding to the upper tail of the distribution of the chosen variable, as the highest values recorded in a certain time period. Moreover, conducting the same analysis on the deseasonalized time series (which is anyway possible within our scheme) would give different results, which answer to a different scientific question. Please see also our reply to the Overall Comments.

To complete the list of our corrections related to the semantic question raised so far by Reviewer#1, we report here our revised expressions about "extreme events", "bloom" and "maxima" (of chlorophyll/of blooms) not explicitly indicated in our previous reply to Reviewer#1's comments posted in the Biogeoscience open discussion.

In particular, we replaced "extremes" with:

*extreme events*

at lines 20, 46, 50, 64, 65, 85, 89, 203, 292, 295, 393, 409, 412-413, 425, 443, we deleted the expression "for brevity: "extremes" at old line 82 and we replaced "the extreme" with:

*extreme event*

at lines 293, 405.

We replaced the title of Sect. 2.1 (i.e., "The method for spatio-temporal extremes investigation") with:

*The method for the spatio-temporal investigation of extreme events*

and the sentences at old lines 156-158 (lines 161-162 in the new version of the manuscript) with:

*In particular, we used the daily chlorophyll concentration computed at the surface (i.e., averaged on the first 10 m), restricting our investigation to the January-May period.*

We added lines 174-175:

*The local "extreme events" for this application thus correspond to the top 1% values of the surface chlorophyll distribution in each grid point (Sect. 2.1.1).*

We modified the expression "inter-annual variability of the blooms" at old line 292 by:

*inter-annual variability of the extreme events of surface chlorophyll*

at line 311.

We replaced "blooms belonging to cluster #7" with:

*EEWs occurring in cluster #7*

at line 344 and "blooms whose chlorophyll values" with:

*extreme events of surface chlorophyll whose values*

at line 357-358.

We replaced at line 372 "a persistence of the blooms" with:

*a persistence of extreme events of chlorophyll*

We modified "In the specific application to the open-sea Mediterranean chlorophyll, we characterised the maxima of "blooms" (in a local and statistical sense)" by:

*In the specific application to surface chlorophyll in the open sea areas of the Mediterranean, we characterised the top 1% values of chlorophyll distribution*

at lines 432-433.

We replaced "blooms" with:

*chlorophyll EEWs*

at lines 434-435, 437, 438.

Apart from this (important) semantic question, the method is nice and prove to be able to identify, characterize and track the EEW beautifully.

– Also, talking about extremes, i wanted to rise a question, just for discussion. I understand the choice of surface Chl maxima is mainly to test the method and show how it works. But thinking about Mediterranean sea, climate change and extreme events, i wonder if tracking maxima of surface chlorophyll maximum is what i would do. I don't think we can get hypoxia or eutrophication with 12th degree model, this is rather a coastal and river mouth problem. We know that a climate impact could be to lower the deep water formation and hence the bloom. We could use your method (adapting the threshold, considering the Annual cycle) to track years with little or no bloom, and understand why, and see the trends. Or in summertime if your model include dust deposition on high frequency, see if the model shows EEWs linked to dust deposition events,... There is lots of other application of your methods that could make lots of sense (Lots of nice study in perspective).

Thank you for this meaningful observations.
As a first application of the method, we chose the surface chlorophyll since: it is representative of the marine ecosystem functioning; it has been widely investigated in previous studies; model simulation is comparable also with remote sensing measurements (as done in Sect. 3.1, Fig. 4), which increases the confidence level on our model-derived results. These reasons make the chlorophyll a good choice to show how the method works, as you wrote.
However, we agree that there are a lot of possible and interesting applications, thanks for your suggestions. We deepened the discussion about the possible applications in Sect. 4, lines 393-402.
As written in the last part of our reply to the Overall Comments, we explicitly added in Discussion section (lines 414-418) the possibility to apply our method starting from seasonally varying threshold, with a mention of the anomalies of chlorophyll in summer as example of investigated process.

== **Text remarks**

– I think there are few places where the English could be corrected, but not being a native English myself, i am not the right person to do that. Maybe you could ask a native English around you to double check your manuscript.
We followed your suggestion and sent the new version of the manuscript to an English Editing Service (whose certificate is added at the end of this document).

– from p5.l134 and all units following : double check the units the -2 and -1 should be up, if you write with latex, you should write kg km^{-2} day^{-1}
Done

– p6 l73 : " chlorophyll as a proxy for the phytoplankton biomass" Surface chlorophyll is representative of the surface biomass (probably why one of your idea in the discussion is to check the event in 3D)
We corrected the expression by:
*surface chlorophyll as a proxy for surface phytoplankton biomass*
at line 178.

– p6 l82 to 85: "Mapping the 99th percentile threshold values computed at each grid point on the whole basin (Fig. 3), it can be noticed that grid points that are near in space exhibit small differences in their threshold values and also that different patterns are recognisable in the basin. Hereafter, we use the abbreviations indicated in Fig. 3 to refer to different Mediterranean regions" – So the 99th percentile is fixed in time. This means that you compare toward an ~annual 99th percentile threshold of Chl. basically you will only have EEW during the bloom period. A summer with exceptional summertime Chl will not appear with this method as it will never exceed bloom period values. can't you do a time varying 99th percentile threshold to be able to see non- bloom period EEW ? otherwise you will probably miss the most interesting events... probably needs a longer run to get enough data to keep it statistically feasible.

We are aware that a percentile threshold which is fixed in time allows to recognise only the highest values of chlorophyll in the considered time period, and in this paper we focus on those events. We do not exclude to investigate also fall blooms, or high anomalies of chlorophyll in summer, but these are different processes, and would deserve separate papers. Please see also our reply to the Overall Comments.

– p7 l191 : " The model-derived chlorophyll patterns (Fig. 4, second column) are in good agreement with the remote sensing data (first column) in the same temporal interval of the EEW" – Hard to tell, seems the sat Chl has a more extended bloom than the model, and starts slightly later (and probably ends later as well). But both model and sat presents an EEW on the same period, what is already a nice model performance ! And you have a nice bloom in the Ligurian sub-basin, that's impressive! Talking about e Ligurian bloom, it does not appear in the EEW area. it is considered as a separated EEW ?

Thank you for this comment. We replaced lines 191-193 "The model-derived chlorophyll patterns (Fig. 4, second column) are in good agreement with the remote sensing data (first column) in the same temporal interval of the EEW. In fact, a strong increase…" with:
*Both the model and satellite data show patterns of high values of chlorophyll in the period of EEW occurrence (second and first columns of Fig. 4, respectively). Strong increases…*
at lines 196-198.
However, the high values recorded in the Ligurian Sea are included in another EEW. We added this information at lines 209-210 as:
*The high values of chlorophyll recorded in the Ligurian Sea on 20th March are associated with a separate EEW (not shown).*

– p8 l235 : " are around half of the ones of ALB or NWM." needs to be rephrase.
We rephrased it (at lines 242-243) as:
*being approximately 50% lower than the values displayed in ALB or NWM*

– p8 l239 : "with a similar chlorophyll EEWs phenomenology."
Thank you for the suggestion. We modified the first part of the sentence that includes this expression (line 247) as:
*Seven Mediterranean Sea regions with similar chlorophyll EEWs phenomenology were identified...*

– p10 l288 : "pointed out the heterogeneity of the blooms intensity in the Mediterranean Sea" - back to my main comment, you don't see blooms everywhere...
As we replied to the Overall Comments, we have avoided the use of the term "bloom" in the new version of the manuscript. We replaced that expression with:
*revealed the heterogeneity of the chlorophyll EEWs in the Mediterranean Sea*
at lines 307-308.

– p10 l310 : Furthermore ?
Thank you for the correction. However, we preferred to delete the whole sentence at old lines 310-311, as reported in the next reply.

– p10 l310 : you could have shown the "spatio-temporal persistence", it looks like a nice index. Why not show it ?

We did not include the map of the spatio-temporal persistence in the first version of the manuscript because this additional index is directly obtained as the product of two indexes, more meaningful in our opinion, which have been already shown in Fig. 5: uniformity, i.e., (spatial) persistency of the EEW, and duration, i.e., total time of occurrence of the EEW.

We recognised that this index should have been defined in Sect. 2.1.2 before its mention in Discussion, but we preferred to avoid it, to not weigh down the manuscript. Therefore, we decided to delete the sentence at old lines 310-311.

– p12 l360 to 374 : Good idea!

Thank you for the comment.

– p12 l375 : "A critical parameter of our method is the choice of the local percentile threshold" - I agree looks like one of the key of the method. but why this choice of a simple percentile threshold, and not include the local annual cycle ( maybe + a*STD ) in the threshold (As i mentioned above depending what you want to analyse, it can be justified, it can be the right choice) ?

We agree that the choice of the threshold depends on the scientific question. Please see our reply to the Overall Comments.

– p13 l 197 : "Of the clusters with the highest content of all the indexes has been generally maintained both in case of higher and lower thresholds" - rephrasing : of the clusters with the highest index values,...,

Done

– p13 l396 : "A key issue" - not issue, it is one of the strength of this method, not issue i think, and from all what you could do in your study because of that.

We agree with your comment. We rephrased "key issue", by using:

*key point*

at line 423.

– p13 l400 : "The time series in the grid point" - rephrase : Each grid point's time serie

Thank you for the suggestion.

The sentence including this expression was corrected by the English Editing Service, but the indicated expression was not modified.

– p13 l400 : " allowed to maintain a definition of "extreme" relative to the local ecosystem properties." This i do not agree. in some places like the most oligotrophic regions, you probably find extremes, but in the bloom regions, it is not.

We do not agree with this observation. Also the time series recorded in points belonging to bloom regions (e.g. Gulf of Lion) can show an inter-annual variability such that the 99th percentile threshold is overcome only in some years, as shown in Fig. R1.2 in the first part of this reply.

–P23-Fig4–You talk about the MLD in the text, but you don't show it on the plot. Of course, we can guess the Mixed layer is very deep where the NO3 is high and Chl low, but, it might be good to add iso-contour with depth values on the Chl plot for example. That would help both the writer and the reader. – very nice bloom in the Legurian sub- basin! You must have a very high res atm model with high freq coupling. you should add these details in the model description. It help understand the results.

In the reviewed version of the manuscript we added further details related to the atmospheric forcing in the model description, at lines 158-160:

*The atmospheric fields used to force the simulation come from a 12 km horizontal resolution regional downscaling of ERA-Interim reanalysis (Llasses et al., 2016; Reale et al., 2017) and drive the simulation every 3 hours.,*

thanks for the suggestion.

With regard to MLD, we have already plotted its time series in three points, internal, peripheral and external to the EEW in Appendix (Figs. A.2-A.4) to help the interpretation of the results.

We recognise that including MLD also in Fig. 4 can help the reader, but we are worried about the readability of the new resulting figure, since Fig. 4 is already full of detailed information. Thus, we decided to keep Fig.4 as it is now and to refer to Figs. A.2-A.4 for MLD evolution.

– P24 - Fig 5a – Difficult to interpret.... and the color-scale does not help. how many EEW occur per year ? how many on the hole period ? I don't understand what you mean here.

We recognise that Fig. 5 in the previous version of the manuscript needed graphic improvement. However, Fig. 5a represented in each grid point the probability of occurrence of more than one EEW per year. However, considering Reviewer#1's comment, we decided to modify it, by representing more simply the total number of EEWs in the period 1994-2012. Therefore, we replaced Fig. 5a with the following (Fig. R1.3):

[Figure]

**Fig. R1.3** - Proposed reviewed Fig. 5a.

and the first part of the Fig.5 caption accordingly:

*Fig.5: Number of surface chlorophyll EEWs that occurred in Mediterranean Sea in 1994-2012 (a) and means of the indexes referring to the EEWs:...*

Moreover, we modified the text referred to Fig. 5a, in Sect. 3.2.1 at lines 230-234, as:

*Figure 5 displays the total number of EEWs that occurred in each point of the Mediterranean domain (Fig.5a) and the mean values of the EEW indexes, which were computed as the mean of the indexes of all the EEWs that involved that point (Figs. 5b-f).*

*Since some Mediterranean areas show more than one EEW per year (as can be inferred from Fig. 5a), the initiation time in each grid point and year was associated with the most severe EEW of that year.*

and in Discussion, at lines 329-330:

*The initiation index was excluded from the computation since there are areas of the basin showing more than one EEW per year per grid point (Fig. 5a).*

– P27 - Fig A.2, A.3, A.4 – I cannot do the difference between the climatological line end the 2005 one. Please, try with different dashed or doted line to find one that we really can see.

Thank you for your comment. We have improved the graphic quality of the Figs. A.2, A.3, A.4 in the new version of the manuscript, as reported here for the new Fig. A.2 (Fig. R1.4). The other figures have been modified accordingly.

[Figure]

**Fig. R1.4** - Proposed reviewed Fig. A.2.

*In fact, we have applied this method to surface chlorophyll, as one of the most representative and investigated variables of the marine ecosystem, which potentially influences ecosystem function (e.g., food web and carbon fluxes). However, our method can be applied to any ecosystem variable, including other phytoplankton variables (e.g., HAB-like phytoplankton groups, Vila and Masó, 2005), temperature, oxygen and fluxes (e.g., carbon fluxes at the ocean-atmosphere interface, von Schuckmann et al., 2018). The C(x,y,t) variable can be defined at the surface, the sea bottom (e.g., oxygen minimum or oxygen deficiency, OSPAR, 2013; Ciavatta et al., 2016), and specific surfaces in the ocean interior (e.g., deep chlorophyll maximum, Lavigne et al., 2015; Salon et al., 2019), or it can be vertically integrated (e.g., integrated chlorophyll, which accounts for subsurface growth of phytoplankton). In some cases, the selected variable may require multiplication by the cell volume (e.g., if the variable is a concentration) or by the cell area (e.g., for surface fluxes and vertically*

*integrated variables) in eq. (2.3) and eq. (2.4) to provide a consistent and meaningful definition of the severity and the excess indexes, respectively.*

This part of the Discussion section refers to the modified definitions of severity and excess indexes (eqs. (2.3) and (2.4)):

$$S = \sum_{(x,y) \in A} \quad M(x,y) = \sum_{(x,y) \in A} \quad \sum_{j \in J_{(x,y)}} \quad C_j(x,y) \tag{2.3}$$

$$S = \sum_{(x,y) \in A} \quad M(x,y) = \sum_{(x,y) \in A} \quad \sum_{j \in J_{(x,y)}} \quad C_j(x,y) \tag{2.4}$$

at lines 127 and 131, respectively, and to the lines 133-135:

*Depending on the ecosystem variable under investigation, eqs. (2.3) and (2.4) may require multiplication by the cell volume or by the cell area in the inner summation to provide a consistent unit of measurement (e.g., if $C_j(x,y)$ is a concentration, it should be multiplied by the cell volume $V(x,y)$ to obtain an actual mass).,*

which were added as a consequence of the extension of the method to any ecosystem variable, declared at line 74:

*with reference to any ecosystem variable C(x,y,z,t)*

Anyway, as we replied also to Reviewer#1, we decided to avoid the reference to "bloom" processes in the new version of the manuscript, and to restrict our argumentation to extreme events (i.e., top 1% values of surface chlorophyll distribution).

3) It is discussed but I am still worried about the stability of the identified regimes using different thresholds. You showed that the clustering of the mean values of the indices do not change much. However, are there changes in the spatial distribution of the regimes shown by Figure 6?

Thank you for this comment, which gives us the opportunity to deepen some aspects.
We applied the fuzzy k-means analysis also in case of 98th and 99.5th percentile thresholds and we report the results in Figs. R2.1 and R2.2, respectively.

[Figure]

**Figure R2.1** Fuzzy clusters with maximum membership, in case of 98th percentile threshold.

[Figure]

**Figure R2.2** Fuzzy clusters with maximum membership, in case of 99.5th percentile threshold.

From a general point of view, the cluster distribution for 99.5th percentile threshold (Fig. R2.2) is very similar to the cluster distribution for 99th percentile (Fig. 6 of the manuscript), while the cluster distribution for 98th percentile (Fig. R2.1) differs mainly in Ionian and Levantine Sea.

More in details, the spatial distribution of clusters #3, #6, #7 in the western basin does not considerably change with respect to Fig. 6 of the manuscript, as well as the spatial distribution of cluster #4 in the eastern basin.
Clusters #1, #2, #3 #5 and #6 display instead differences, mainly in the cluster distribution for 98th percentile, in the eastern basin, i.e., southern Adriatic Sea, central Ionian Sea, Aegean Sea, Libyan coast and eastern Levantine Sea.

Since the identification of clusters depends on four indexes (i.e., mean severity, uniformity, duration, anomaly) which do not necessarily scale in the same way with the local threshold, the fact that the spatial distribution resulting from the fuzzy k-means analysis (as "combination" of the four indexes) for different thresholds can differ from Fig. 6 is a reasonable result.
Moreover, it should be highlighted that Figs. 6, R2.1, R2.2 show the clusters referred to the maximum membership.
In addition to maximum membership, we also considered the "confusion index" (Burrough et al., 1997), which, applied to our case, quantifies how well each point of the Mediterranean domain has been classified. High values of the confusion index (CI) index are related to higher sensitivity of some areas in the cluster classification with respect to variations in the local threshold (i.e., to differences of Figs. R2.1, R2.2 with respect to Fig. 6).
We estimated the confusion index (CI) as:
$CI=1-(MF_{max}-MF_{max2})$,
where $MF_{max}$ denotes the dominant membership value and $MF_{max2}$ is the subdominant membership value for each point, and we computed it in case of 99th percentile threshold. Figure R2.3 shows values of CI greater than 0.7 (i.e., "high values" of CI) as black dotted points.

[Figure]

**Figure R2.3** Fuzzy clusters with maximum membership, in case of 99th percentile threshold, with black points indicating a confusion index higher than 0.7.

We can observe that most of the areas displaying differences in the spatial distribution of the clusters with respect to variations in the threshold (Figs. R2.1, R2.2) correspond to high CI in the reference case (Fig. R2.3) and that the identification of the clusters generally appears to be consistent with the other two clusterizations.

We recognised that this point deserved a revision in the manuscript.
Therefore, we replaced Fig. 6 by Fig. R2.3 (updated to the format requested by the journal) in the revised manuscript, adding the expression:
*with black points indicating a confusion index higher than 0.7*
at the end of the figure caption and the definition of the confusion index in the text. Accordingly, our statement about the robustness of the classification (old lines 387-390) have been reformulated and referred to this new version of Fig. 6.
In particular, we replaced old line 239 by:
*Seven Mediterranean Sea regions with similar chlorophyll EEWs phenomenology were identified by the maximum membership values and are indicated by different colours in Fig. 6. To evaluate the robustness of the clusterisation, we also computed the "confusion index", i.e., one minus the difference between the dominant and subdominant memberships for each point (Burrough et al., 1997). We obtained a value less than 0.7 (i.e., limit for "high confusion" condition) for the largest part of the domain. Values higher than 0.7 are shown in only patchy and limited areas (e.g., part of the southern Adriatic Sea, Fig. 6). Moreover, we computed the mean...*
at lines 247-252.

Moreover, we replaced old lines 318-320 by:
*We obtained robust clusterisation, with only some areas of the domain showing a high confusion index (Fig. 6). This subdivision of the Mediterranean Sea displays several similarities to previous Mediterranean bio-regionalisations (D'Ortenzio and Ribera D'Alcalà, 2009; Lazzari et al., 2012;*

*Ayata et al., 2018; Salon et al., 2019), indicating that the four indexes are meaningful in characterising the heterogeneity of the basin.*
at lines 334-337.
We replaced old lines 387-390 by:
*Overall, Fig. 7 shows that the identification of the clusters with the highest index values was generally maintained in the case of both higher and lower thresholds, confirming that the main regimes of chlorophyll EEW were identified in a robust way.*
at lines 390-391.
Finally, we further clarified the text at old lines 263-265, that we reformulated at lines 275-278 as:
*We repeated the steps of the method (Sects. 2.1.1-2.1.3) up to obtaining the mean maps of the indexes on the Mediterranean domain for the 98th and 99.5th percentile thresholds. Then, we spatially averaged the values of the indexes within the seven clusters of Fig. 6 and finally we computed the total means by averaging the means of the seven clusters.*

4) In Section 4 and the Appendix, ecosystem dynamics characterizing some of the regimes are discussed. However, some clusters lack any dynamical explanation and might be rather artificial. It would increase the scientific value of the manuscript if you could discuss these clusters more in detail as you have done it for NWM in the Appendix.

Section 3 and Appendix of the first version of the manuscript present in detail one of the extreme event waves (EEWs) identified in the 1994-2012 dataset, to show how the method works. In particular, Appendix highlighted that the method catches all and only the relevant information of the event. We selected the EEW associated to the highest value of mean severity of the whole dataset (and occurred within the area covered by cluster #7) as a particularly meaningful case.
Since the main focus of the paper is to propose a method to identify and classify EEWs, rather than to analyse in detail surface chlorophyll dynamics, we did not include other examples of EEWs (e.g., associated to areas covered by other clusters, as suggested) in the first version of the manuscript, avoiding to enlarge too much the length of the manuscript and to shift the attention more on the specific application than on the method. For the same reasons, we decided  not to add new figures and comments in Appendix of the revised manuscript.
However, if the Reviewer#2 still suggests to do it, we are open to include other examples of EEWs occurring in areas covered by other clusters.

5) In the abstract you mentioned that "There is a growing interest about events that can affect ecosystem functions and services in a changing climate". However, is the method suitable for following the temporal shifts in the regimes without any discontinuities? I suggest that you split the period in half and that you use the cluster analysis for both periods. By this, the impact of trends in some of the indices on the spatial distribution of the regimes might be investigated and compared with observed changes.

Thank you for the comment.
Yes, our method is suitable for following the temporal changes in the characteristics of the EEWs, provided that the time series is long enough for a robust trend or regime shift evaluation on a multi-decadal scale. Unfortunately, our time series is not very long (i.e., 19 years). Thus, we think that the approach to split the chlorophyll dataset in two parts, i.e. 9-10 years for each part, would not guarantee statistical robustness to the analysis.

Indeed, we conducted a non parametric analysis of the trend slope (Theil-Sen method) and showed the results in Tab. 2 of the manuscript, by reporting the cases in which the ratio between slope and mean is higher than 1% (red cells) and lower than -1% (blue cells). For sake of clarity, we report here the annual time series (and trend slope) of each cluster for the duration, uniformity, mean severity and anomaly indexes (Figs. R2.4-R2.7). As it can be intuitively grasped, the length of the time series does not allow further speculations on temporal changes of the EEWs characterisation and classification.

[Figure]

**Figure R2.4** Trend evaluation (as slope over mean) on the clusters in Fig.6, referred to duration index.

[Figure]

**Figure R2.5** As Fig. R2.4, but referred to uniformity index.

[Figure]

**Figure R2.6** As Fig. R2.4, but referred to mean severity index.

[Figure]

**Figure R2.7** As Fig. R2.4, but referred to anomaly index.

Bibliography:

[revised manuscript text omitted]

**SPRINGER NATURE**
**Author Services**

**Editing Certificate**

This document certifies that the manuscript

**Paper on Extreme Events Waves by Di Biagio et al.**

prepared by the authors

**Valeria Di Biagio, Gianpiero Cossarini, Stefano Salon, Cosimo Solidoro**

was edited for proper English language, grammar, punctuation, spelling, and overall style
by one or more of the highly qualified native English speaking editors at SNAS.

This certificate was issued on **July 7, 2020** and may be verified
on the SNAS website using the verification code **C1D7-8A0C-062A-5BBE-7600** .

Neither the research content nor the authors' intentions were altered in any way during the editing process. Documents receiving this certification should be English-ready for publication; however, the author has the ability to accept or reject our suggestions and changes. To verify the final SNAS edited version, please visit our verification page at secure.authorservices.springernature.com/certificate/verify. If you have any questions or concerns about this edited document, please contact SNAS at support@as.springernature.com.

SNAS provides a range of editing, translation, and manuscript services for researchers and publishers around the world.
For more information about our company, services, and partner discounts, please visit authorservices.springernature.com.

---

## Author Response (AR2)

Dear Marilaure Grégoire (editor), we thank you and the two reviewers for assessing our manuscript and for the time and effort dedicated to it.
Please find below our reply to the latest comments by Reviewer #1, including a list of the changes made in the manuscript, and our revised manuscript, with modified text highlighted in red.

Following your suggestion and Reviewer #1's comments, we added in the manuscript another appendix (i.e., Appendix A, whereas the previous appendix has been defined Appendix B), displaying extreme events of surface chlorophyll in selected sites of the Mediterranean Sea (that are indicated in the new version of Fig. 3).
Figures A.1-A.3 show that not all the highest annual values of chlorophyll (corresponding to "blooms" in the north-western Mediterranean case) correspond to extreme events. Moreover, the same figures highlight features of intensity and inter-annual variability of the extreme events that are different in the different Mediterranean sites.

We hope that this revised version is suitable for publication in Biogeosciences.

Kind regards,

Valeria Di Biagio on behalf of the authors
* * *
**Reply to Reviewer#1's comments**

We sincerely thank Reviewer#1 for his/her comments, which gave us the opportunity to clarify some key points and significantly improve our paper.
We indicate our reply in blue colour and the corrections implemented on the text of the new version of the manuscript in italic red.

**Suggestions for revision or reasons for rejection (will be published if the paper is accepted for final publication)**

I recommend the publication of the paper.
Last and least suggestion (why i said technical correction), is to ask Valeria Di Biagio to add the picture she mentions p8 of her answer to referees, as this picture (like R1.1 and R1.2) will help convincing biologist like me that not all N-W Mediterranean blooms are extreme events, and i think it gives a nice overall picture of "how extreme are the EEWs" in the different regions of the Mediterranean sea, giving a nice illustration of the story told by the EEW's indexes.
Other than that, the paper is ready for publication!

We thank Reviewer# 1 for his/her positive evaluation of the manuscript.

We followed his/her suggestion to add in the new version of the manuscript figures similar to R1.1 and R1.2 included in our previous reply to reviewers.
In fact, we added a new appendix (namely, Appendix A), showing the identification of extreme events of surface chlorophyll in selected sites belonging to the different Mediterranean Sea regions (these sites are indicated in the new version of Fig. 3).
Figures A.1-A.3 show that not all the highest annual values of chlorophyll (including the north-western Mediterranean "blooms") correspond to extreme events. Moreover, these figures highlight the different features of intensity and inter-annual variability of the extreme events displayed by the selected sites.
Furthermore, it is worth to highlight that quantifying how "extreme" are the EEWs in the different Mediterranean areas is instead done from Fig. 5 and Table 2 of the manuscript.

The following text of Appendix A was added at lines 448-454:

*Appendix A: extreme events of surface chlorophyll in Mediterranean Sea regions*

*Figures A.1-A.3 show time series and distribution of surface chlorophyll in selected sites of the Mediterranean Sea (Fig. 3).*
*The extreme events are identified by the surface chlorophyll values higher than 99th percentile threshold computed in the sites, i.e., by the upper tail (top 1%) of the corresponding distribution. The selected sites show different 99th percentile thresholds, as well as different values and temporal occurrence of the local extreme events. Despite in each site the highest annual values of surface chlorophyll well exceed the maximum values of the climatological seasonal cycles, they do not necessarily correspond to extreme events. Extreme events are in fact identified in some years only.*

Figures A.1-A-3, with captions indicated below, were added at lines 747-756:

[Figure]

***Figure A.1***: *Left panels: time series of model-derived surface chlorophyll (green series) and daily climatology (black series) computed in ALB, SWW and SWE sites indicated in Fig. 3; right panels: model-derived surface chlorophyll distribution in the corresponding sites, with x axis in logarithmic scale. Red dashed lines represent the local 99th percentile thresholds in all panels.*

[Figure]

**Figure A.2:** *As in Fig. A.1, but referred to NWM, TYR and ION sites, respectively.*

[Figure]

**Figure A.3:** *As in Fig. A.1, but referred to ADS, AEG and LEV sites, respectively.*

We report here also the new version of Fig. 3, displaying the selected sites, belonging to different Mediterranean Sea regions:

and in the caption of this figure we added (lines 719-720):
*Black squares indicate sites chosen as examples to display extreme events of surface chlorophyll in the Mediterranean regions (Appendix A).*

Moreover, we modified the name of the previous Appendix in Appendix B and the name of Figs. A.1, A.2, A.3 and A.4 included in the previous version of the manuscript in B.1, B.2, B.3 and B.4, respectively, throughout the text of the manuscript (i.e., lines 69, 220, 316, 455, 456, 457, 462, 473, 475, 762, 767, 769, 774, 778).

We added the reference to the Appendix A in Introduction (lines 66-67) as:
*The results are presented in Sect. 3 and Appendix A (examples of identification of extreme events of surface chlorophyll). In particular, Sect. 3.1 shows a chlorophyll EEW ...*

In Sect. 3.1, we added at lines 176-178:
*Examples of extreme events of surface chlorophyll in Mediterranean Sea are reported for selected sites (Fig. 3) in Appendix A, showing that the different sites have different p99 and that not all annual maxima correspond to the definition of local "extreme events".*

Finally, in Discussion we added a reference to the Fig. A.2, that shows the identification of extreme events in the selected ION site, as (lines 321-322):

[revised manuscript text omitted]

---

## Author Response (AR3)

Dear Marilaure Grégoire (editor),

Thank you for accepting the paper.
We uploaded the final version of the manuscript, with just a few modifications on the style of figures A.1, A.2, A.3, in order to adopt the same format in all the figures.

Best regards,

Valeria Di Biagio on behalf of the authors